# HCMV encoded UL84 hijacks FHL2 to suppress type I interferon production and enhance viral replication

Ruilin Li[1☉], Sisi Xia[2☉], Xin Li[1☉], Ying Zeng[3], Tianqi Wang[3], Chuan Xia[3]*, Hongjian Li[1]*, Jun Chen [1]*

1 State Key Laboratory of Bioactive Molecules and Druggability Assessment, Guangdong Basic Research Center of Excellence for Natural Bioactive Molecules and Discovery of Innovative Drugs, Key Laboratory of Viral Pathogenesis & Infection Prevention and Control, College of Life Science and Technology, College of Pharmacy, Jinan University, Guangzhou, China, 2 Department of Biological Engineering, Wuhan Polytechnic University, Wuhan, China, 3 Department of Pathogen Biology and Microecology, College of Basic Medical Sciences, Dalian Medical University, Dalian, China

☉ These authors contributed equally to this work.
* chenjun@jnu.edu.cn (JC); tlihj@jnu.edu.cn (HL); xiachuan@dmu.edu.cn (CX)

## Abstract

Virus infection activates the host's innate immune responses, which is a very precise and complex biological process and will lead to the immediate transcription of type I interferon. The general transcriptional activator proteins such as IRF3, ATF2/c-Jun, and NF-κB can be induced to form a stable enhanceosome in the transcriptional regulatory region of IFN-β promoter. Several cellular factors have recently been reported to be involved in the transcriptional regulation of IFN-β under certain physiological conditions. Here, we identified four and a half LIM domains protein 2 (FHL2) as an interacting protein of the Human Cytomegalovirus (HCMV) replication-related protein UL84 and determined that FHL2 plays an architectural role in enhancing the transcription of IFN-β induced by HCMV infection and many other viruses. Firstly, after the virus binds to the host cell, the signal is transmitted to protein kinases, causing the cytoplasmic FHL2 to be phosphorylated and translocated into the nucleus. Then, the phosphorylated FHL2 promotes the formation of the transcription preinitiation complex (PIC) of the IFN-β promoter. Simultaneously, FHL2 is also crucial for the recruitment of TFIID to the TATA-box for initial transcription. Interestingly, during HCMV infection, HCMV replication-related protein UL84 was determined to interact with FHL2 to help the virus evade innate immune response and promote viral lytic origin (oriLyt) dependent DNA replication. Our results highlight the FHL2 as part of a signaling cascade during viral invasion, and its important regulatory effect in type I interferon synthesis, as well as provide theoretical support for the development of candidate anti-HCMV drugs acting specifically on a novel UL84 target.

**Data availability statement:** The date that support the findings of this study are openly available at Mendeley (https://data.mendeley.com/datasets/w2jm2dkv3f/1) DOI: 10.17632/w2jm2dkv3f.1).

**Funding:** This research has been supported by grants from the National Natural Science Foundation of China (grant nos. 32370150 to JC, 32270138 to HL, 32070149 to HL) and Liaoning Provincial Science and Technology Joint Program (LJ212510161001 to CX) and the Fundamental Research Funds for the Central Universities (21625411 to JC) and Science and Technology Project in Guangzhou (grant number 202102070001 to JC). The funders had no role in study design, data collection and analysis, decision to publish, or preparation of the manuscript.

**Competing interests:** The authors have declared that no competing interests exist.

## Author summary

HCMV-encoded UL84 was originally identified as an essential protein for viral lytic origin (*ori*Lyt)-dependent DNA replication. In this study, we provided the first evidence that UL84, as a dual-functional protein, can also suppress host innate immune responses. Mechanistically, UL84 inhibits the transcription of IFN-β by competitively binding to FHL2, which disrupts the assembly of the transcription preinitiation complex and the recruitment of TBP. Meanwhile, we found that phosphorylation of FHL2 acts as part of a signaling cascade after virus-triggered and facilitates the formation of the IFN-β enhanceosome complex. The study highlights the novel function of HCMV-encoded UL84 in innate immune regulation and FHL2 as part of a signaling cascade during viral invasion, as well as its important regulatory effect in type I interferon synthesis. Therefore, candidate drugs targeting UL84 may not only inhibit HCMV replication but also enhance host immune responses.

## Introduction

As the first line of host defense against the viral infection, innate immune responses are triggered in virus-infected cell following the binding of viral ligands or pathogen-associated molecular patterns (PAMPs) to pattern recognition receptors (PRRs) [1,2]. The primary subsequent antiviral pipeline of innate immunity is the production of type I interferon (IFN-I, mainly IFN-β), which can stimulate the expression of more than 2,000 immune related genes [3,4]. During viral infection, a variety of transcription factors, including IRF-3/7 (interferon regulatory factor 3/7), ATF-2/ c-Jun, and NF-κB, are activated to regulate the expression of type I interferon independently or in cross-talking. Among these factors, the host IRF-3 plays a critical role in IFN-β transcription activation [5].

IRF3 is phosphorylated by TBK-1/IKK-ε kinases in the virus-triggered innate immune signaling pathway, and then translocated to the nucleus as functional dimers [5–7]. ATF2/c-Jun is originally present in the nucleus and activated by phosphorylation of the C-terminal amino acid residue by MAP kinase [8]. Following viral infection, the phosphorylation of IκB releases the binding of NF-κB, which is then translocated into the nucleus [9]. These transcription factors bind to different regions of the IFN-β promoter, allowing interferon transcription to be regulated in multiple levels. Moreover, IRF3, ATF2/c-Jun, and NF-κB proteins can be induced to form a basic enhanceosome in the transcriptional regulatory region of IFN-β promoter [10]. Specifically, the general architectural transcription factor HMG I (Y) bends DNA to recruit ATF2/c-Jun and NF-κB, which then assemble with IRF3 to form the enhanceosome complex [7,11]. Then, the enhanceosome complex successively recruits the CREB binding proteins (CBP/p300) and histone acetyltransferases (HATs) to catalyze +1 nucleosome acetylation and promotes the binding of chromatin remodeling complex SWI/SNF to induce TATA-box exposure [11,12]. It then assembles into the preinitiation

complex (PIC) to facilitate the recruitment of TFIID to TATA-box, and rapidly promotes the transcription of IFN-β [13]. However, no direct interaction between IRF3, ATF2/c-Jun, and NF-κB has been found in previous studies, and the specific mechanism of the formation of the basic transcriptional enhanceosome still needs further investigation.

In addition to the general transcription factors and co-factors mentioned above, a few of cellular factors have recently been reported to be involved in the transcriptional regulation of IFN-β under specific physiological conditions. For example, upon virus stimulation, TET3 (Ten-eleven translocation protein 3) interacts with HDAC1 and SIN3A, thus enhancing their binding to the IFN-β promoter, and negatively regulates type I IFN production independent of its DNA demethylation activity [14]. When macrophages are infected with vesicular stomatitis virus (VSV), nuclear factor of activated T cells 5 (NFAT5) inhibits the recruitment of IRF3 to suppress IFN-β transcription [15]. These studies indicate that the transcriptional regulation of IFN-β promoter activity is a dynamic process stimulated by specific extracellular physiological factors.

Four and a half LIM domains protein 2 (FHL2) is a highly conserved and widely distributed protein consisting of four and a half LIM domains, which refer to the cysteine-rich domain originally identified in transcription factors MEC-3 and LIN-11 of *Caenorhabditis elegans*, and in insulin gene enhancer binding protein-1(ISL1) of mouse, and named using the initials of these three proteins [16,17]. Although it lacks any obvious enzymatic activity, FHL2 is a multifunctional protein that shuttles between the nucleus and cytoplasm. The LIM domains, which contain double zinc finger motifs, endow FHL2 with a central function to interact with more than 50 different proteins [18,19]. Many of these interacting partners are cellular receptors and structural proteins. Therefore, FHL2 is considered a scaffold protein that participates in regulating cytoarchitecture, cell adhesion, and cell mobility. However, numerous studies have also suggested that FHL2 functions as a transcriptional regulator, enhancing the transcriptional activity of androgen receptor (AR), c-Jun, AP-1, Egf, Egfr, and TCF/LEF [20,21], as well as regulating the transcription of transforming growth factor β (TGF-β) and the promyelocytic leukemia zinc finger protein (PLZF)-mediated transcription [22,23]. More recently, studies have shown that FHL2 cooperates with IRF3 to upregulate interferon-β (IFN-β) expression, thereby inhibiting influenza virus replication [24]. It can also activate the inflammatory response and reduce the apoptotic events caused by respiratory infection of influenza viruses [25]. From these studies, we can find that the function of FHL2 is always closely associated with IRF3, ATF2/ C-Jun and p300, indicating FHL2 might be involved in the regulation of type I interferon mediated host innate immune response. However, the specific molecular mechanism by which FHL2 exerts its roles still needs to be explored.

Human cytomegalovirus (HCMV) belongs to the β-herpesvirus subfamily and is a widespread pathogen that can cause severe damage to people with low immunity [26]. HCMV has the largest genome in the Herpesviridae family and encodes more than 150 viral proteins, the mechanisms and functions of many viral proteins have not been yet completely elucidated [26]. The HCMV-encoded UL84 protein was originally identified as a non-core replication-related protein essential for the initiation of HCMV lytic DNA replication [27]. Studies demonstrate that during the early phases of HCMV DNA replication, an interaction between UL84 and the origin of lytic replication (oriLyt) is observed [28]. One mode of this binding involves UL84 specifically recognizing and associating with a complex RNA/DNA hybrid structure, which is a defined component of the lytic origin essential for the initiation of DNA replication [28]. Another mode is that UL84 interacts with regions of oriLyt that contain CAAT enhancer-binding protein alpha (C/EBPα) binding sites, thereby promoting viral DNA replication [29]. Additionally, UL84 interacts with the core viral replication proteins UL44 and other replication-related protein IE2, which are necessary for the start of efficient lytic DNA replication [30,31]. However, as one of the non-core proteins essential for DNA-dependent replication, the mRNA transcription of UL84 can be detected as early as 2.5 hours post-infection (hpi), which is considerably earlier than the onset of viral DNA replication [27]. Analysis of the UL84 structure indicates the presence of amino acid motifs that are homologous to those identified in DExD/H family proteins [32]. These motifs may endow UL84 with the capacity for nuclear-cytoplasmic shuttling [32]. Furthermore, luciferase reporter assays demonstrated that the presence of UL84 significantly relieved IE2-mediated transcriptional repression [30]. These findings indicate that UL84 may not only function as a protein related to replication initiation but also participate in the regulation of other viral life-cycle activities.

In this study, we first report UL84 can also inhibit the transcription of IFN-β. Further, we performed a high-throughput yeast two-hybrid screening and identified FHL2 as a binding partner for UL84, and the interaction is confirmed by co-immunoprecipitation, pull-down and confocal microscopy. The UL84-FHL2 interaction is further determined to be involved in the type I interferon transcriptional regulation. HCMV infection of the host activates the innate immune responses, and Type I interferon is a key limiting factor for viral replication and infection. Therefore, we further investigated the specific mechanism by which FHL2 participates in enhancing type I interferon transcription, and how UL84 facilitates the *ori*lyt-dependent DNA synthesis by inhibiting IFN-β transcription.

## Results

### UL84 inhibits HCMV induced type I interferon response

The HCMV genome contains over 200 open reading frames (ORFs), encoding a large number of viral proteins [32]. To identify HCMV proteins that relate to immune regulation, we constructed a series of viral protein expression vectors and screened them using reporter assays for their abilities to regulate IFN-β promoter activity in HCMV-stimulated HFF cells. It's intriguing to find out that the viral protein UL84 markedly inhibits the HCMV-induced activation of IFN-β transcription, despite previous reports have considered UL84 as an origin-binding protein (OBP) for viral replication and an essential protein for this process [32]. As shown in Fig 1A, the luciferase reporter assay showed that UL84 dramatically repressed the transcriptional activity of the IFN-β promoter in a dose-dependent manner. Meanwhile, the analysis of gene expression at different time points also demonstrated that the overexpression of UL84 could continuously inhibit the activity of the interferon-β promoter under the condition of viral infection (Fig 1B).

It has been shown that significant upregulation of IFN-β mRNA is detectable in HCMV-infected HFFs at 4 hpi, while UL84 transcripts are detectable as early as 2.5 hpi post-infection [27,33]. To confirm the temporal expression profile of UL84, the expression levels of UL84 from 3 to 12 hours after HCMV infection were examined. The results indicated that at 3 hpi, a significant upregulation of UL84 transcripts was observed by RT-qPCR, which was consistent with the detection of UL84 protein levels via Western blotting (S1 Fig). HCMV infection of HFF cells could induce the production of various host antiviral interferon-stimulated genes (ISGs) [34]. RT-qPCR analysis indicated that UL84 not only inhibits the transcription of *IFNB1*, but also that of the downstream typical representative genes *IFIT1* and *ISG15* in HFFs (Fig 1C). Next, we designed and screened two small interfering RNAs (siRNAs) that specifically target UL84 and verified their interference effects via Western blot analysis (S3A Fig). RT-qPCR analysis showed that knockdown of UL84 promoted HCMV induced transcription of *IFNB1*, *ISG15*, and *IFIT1* genes in HFF (Fig 1D and 1E).

Interestingly, we discovered that the antiviral genes induced by IAV, EV71, HSV, and ZIKV upon the infection of HFF cells could also be suppressed by UL84 (Fig 1F–1I). In particular, we performed additional experiments to investigate the effect in cells transfected with UL84 on the titer of IAV. In S2A Fig, the data indicate that cells overexpressing UL84 indeed show higher titers of IAV. Furthermore, to verify whether UL84 specifically inhibits interferon responses, we also tested the effect of UL84 on inflammatory-related factors. The results showed that overexpression of UL84 in HFF cells failed to suppress NF-κB promoter activity upon stimulation with HCMV, IAV, EV71, HSV, or ZIKV (S2B–S2F Fig). Furthermore, subsequent RT-qPCR analysis showed that overexpression of UL84 also did not cause significant changes in TNF-α, IL-6 and IRF9 mRNA levels (S2G–S2U Fig).

These results suggest that UL84 can generally inhibit the host IFN-β mediated-antiviral response against different viruses.

### UL84 specifically interacts with host factor FHL2 to subvert host immune defense

To investigate the specific mechanisms of UL84 in immune escape, we performed a high-throughput yeast two-hybrid (YTH) screening to identify cellular factors that potentially interact with UL84, and some of the positive results were

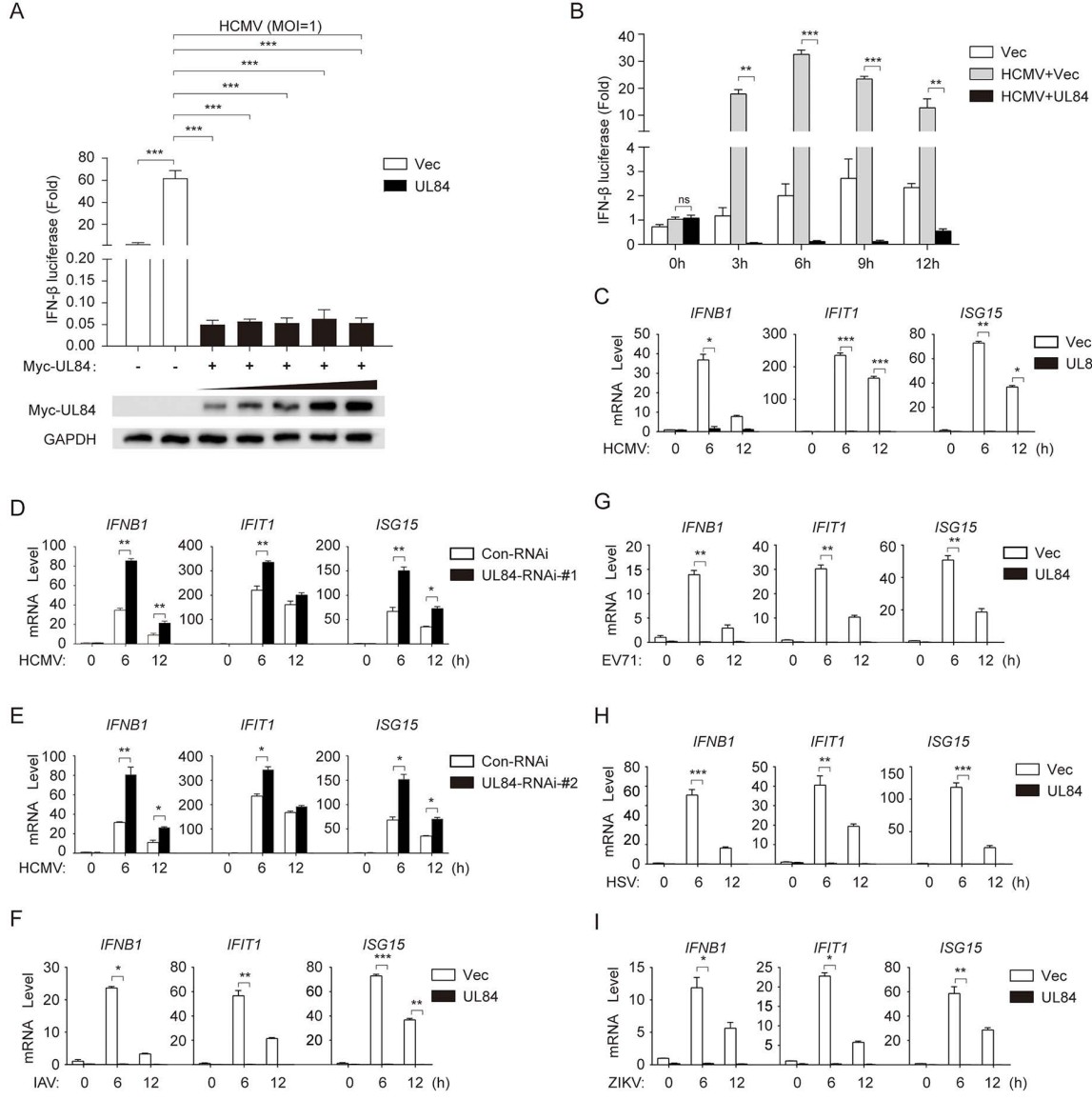

**Fig 1. Identification of HCMV UL84 as an antagonist for IFN-β response. (A)** HFF cells were transiently transfected vector via electroporation with pGL3-luci-IFN-beta and pRL-TK reporter plasmids, along with increasing amounts of the pcDNA3.1-Myc-UL84 expression vector (*indicated as UL84*) or the empty pcDNA3.1 vector as a control (*Vec*) for 24 h. The cells were infected with HCMV (MOI = 1) for 24 h before the dual luciferase reporter assay. **(B)** HFF cells were transiently transfected plasmids pGL3-luci-IFN-beta and pRL-TK together with either pcDNA3.1-Myc-UL84 or the empty vector via electroporation, and then infected with HCMV (MOI = 1) for the indicated times before conducting the dual-luciferase reporter assay. **(C)** HFF cells were transiently transfected plasmid pcDNA3.1-Myc-UL84 or the empty vector via electroporation, and then infected with HCMV (MOI = 1) for the indicated times before reverse transcription real-time quantitative PCR (RT-qPCR) analysis of the indicated antiviral interferon-stimulated genes (*IFIT1*, *ISG15* and *IFNB1*). **(D) (E)** HFF cells were transiently transfected UL84-RNAi-#1 and UL84-RNAi-#2 via transfect, and then infected with HCMV (MOI = 1) for the indicated times before reverse transcription real-time quantitative PCR (RT-qPCR) analysis of the indicated antiviral interferon-stimulated genes (*IFIT1*, *ISG15* and *IFNB1*).**(F) (G) (H) (I)** HFF cells were transiently transfected plasmid pcDNA3.1-Myc-UL84 or the empty vector via electroporation, then respectively infected with influenza A virus (IAV) (MOI = 0.5), enterovirus 71 (EV71) (MOI = 1), herpes simplex virus (HSV) (MOI = 1), or Zika virus (ZIKV) (MOI = 1) for the indicated times before RT-qPCR analysis of the indicated interferon-stimulated genes. All experimental assays were conducted in triplicate with independent biological replicates. For all figures, statistical analyses were performed using two-tailed *t*-test. Differences were considered statistically significant when * denoted $p < 0.05$, ** denoted $p < 0.01$, *** denoted $p < 0.001$, and **** denoted $p < 0.0001$.

further validated by Co-IP (S4 Fig). Among these candidate host factors, the most frequently screened one that attracted our attention was FHL2, whose fundamental function is involved in transcriptional regulation and signal transduction (S1 Table). To validate the interaction between UL84 and FHL2 identified from YTH screening, co-immunoprecipitation (Co-IP) assays were first conducted to confirm this interaction in human cells. To rule out nonspecific interactions and consider the functional context of FHL2, we selected expression plasmids of the host transcriptional regulator EZH2 and the HCMV-encoded UL44 stored in our laboratory as negative controls [35,36]. Myc-UL84 was transiently co-transfected with Flag-FHL2 into 293T cells, while co-expression of Flag-EZH2 or Myc-UL44 was respectively set as a negative control. Subsequently, the cell lysates were incubated with immobilized anti-Myc or anti-Flag agarose beads to isolate the corresponding Myc/Flag-tagged protein complexes. Finally, these isolated protein complexes were analyzed by Western blot using anti-Myc and anti-Flag antibodies. The results indicate specific co-immunoprecipitation between Myc-UL84 and Flag-FHL2, but no significant binding was observed between Myc-UL84 and Flag-EZH2 or between Myc-UL44 and Flag-FHL2 in the control groups (Fig 2A). To further determine whether the interaction between UL84 and FHL2 is direct, namely independent of any involvement from other host cellular or HCMV-encoded proteins, we expressed and purified His-UL84, GST-FHL2, His-UL44 and GST-EZH2 in prokaryotic cells respectively, and analyzed their interaction by GST/His pull-down assay. Similarly, only His-UL84 could mutually co-precipitate with GST-FHL2 (Fig 2B).

Next, the co-localization of UL84 and FHL2 within cellular compartments was visualized via confocal microscopy. First, pcDNA3.1-UL84 and/or pcDNA3.1-FHL2 were transfected into HFF cells for 48 h, followed by the observation of subcellular localization of overexpressed proteins using indirect immunofluorescence assay (IFA) via confocal microscopy. The results showed that UL84 exhibited a uniform distribution in both the cytoplasm and nucleus, whereas FHL2 was predominantly localized to the cytoplasm (Fig 2C). However, upon co-transfection with UL84, FHL2 was observed to colocalize with UL84 in both the cytoplasm and nucleus of cells. Further experiments were conducted to determine whether virus-expressed UL84 could specifically interact with endogenously expressed FHL2 in host cells during HCMV infection. The results indicate that UL84 is also capable of specifically interacting with FHL2 under conditions of cellular infection (Fig 2D, Lanes 8 and 10).

To further investigate the essential binding regions between UL84 and FHL2, a series of truncated mutants of UL84 and FHL2 were constructed. As depicted in Fig 2E, FHL2 consists of four half LIM domains. According to previous reports, even the 1/2 LIM can participate in the interaction with splicing factors [37]. Therefore, we designed the truncated expression vectors based on the division of LIM domains (Fig 2E). Subsequently, the truncations of FHL2 were co-expressed with UL84 in HEK293T cells, and the binding region of FHL2 with UL84 was determined by Co-IP. The results showed that the LIM2 domain (95–158 amino acids (aa)) played a pivotal role in mediating the interaction between FHL2 and UL84, as truncation or deletion of this domain significantly impaired their binding affinity (Fig 2F, Lanes 6 and 7). In contrast, removal of other LIM domains did not appear to affect the interaction.

HCMV UL84 is a core protein that plays an essential role in HCMV replication and comprises 587 amino acid residues [38]. The N-terminal domain of UL84 (1–400 aa) has been reported to interact with multiple host and viral proteins, and thus may be mainly related to the formation of oriLyt-dependent replication initiation complex [30,31]. Therefore, we proposed that the region of interaction between FHL2 and UL84 was most likely situated within the C-terminal domain (400–587 aa). Similarly, based on the structural information reported so far, a series of truncated mutants of UL84 were constructed (Fig 2G). Then, the truncations of UL84 were co-expressed with FHL2 in HEK293T cells, and the binding region of UL84 with FHL2 was determined by Co-IP. As we expected, there was no significant binding or co-precipitation between UL84 (1–399) and FHL2 (Fig 2H, Lane 4). The results showed that the C-terminal domain of UL84 (400–460 aa) was indispensable for the interaction with FHL2 (Fig 2H, Lane 7). Finally, molecular docking simulations were performed to provide supporting evidence of the binding sites or domains of UL84 and FHL2. The simulation results demonstrated that UL84 primarily interacts with residues 67–113 within the FHL2 protein, while FHL2 selectively targets residues 446–731 of the UL84 protein, which are consistent with the CO-IP results obtained from truncated mutants (Figs 2I and S6A).

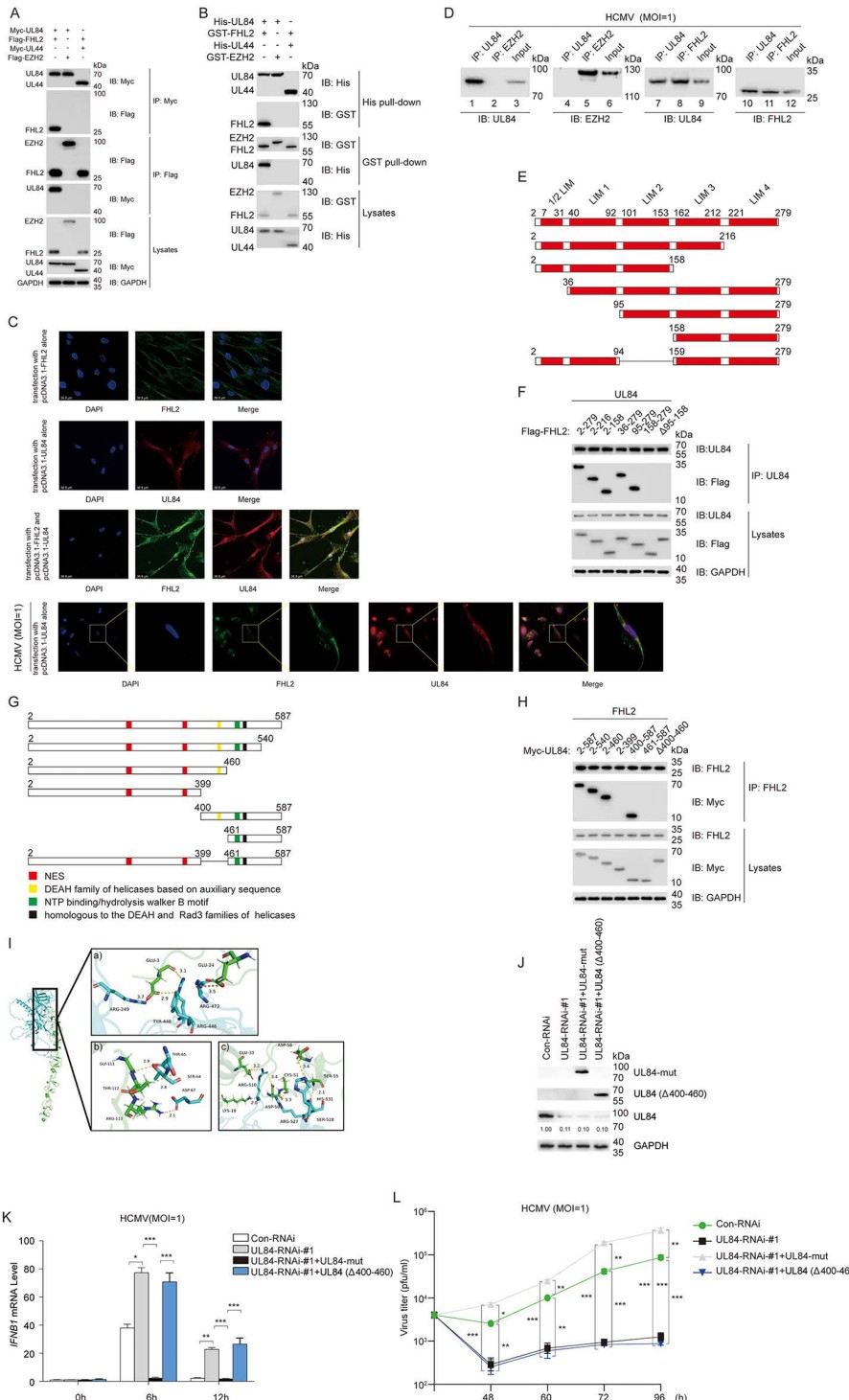

**Fig 2. Confirming and profiling the interaction between UL84 and FHL2. (A)** Co-IP assays between UL84 and FHL2 HEK293T cells were transiently co-transfected with Myc-UL84 and Flag-FHL2 or control plasmids as indicated for 48 h before Co-IP and immunoblots analysis. Myc-UL44 and Flag-EZH2 were set as negative controls to rule out non-specific binding. **(B)** GST/His pull-down assays between UL84 and FHL2. *E. coli* BL21 were transformed with pGEX-6P-1-FHL2, pGEX-6P-1-EZH2, pET28a-UL84, pET28a-UL44 respectively, and then induced with 0.1 mM IPTG for 8 h. His-tagged UL84 or UL44 and GST-tagged FHL2 or EZH2 were individually purified by affinity chromatography using Ni²⁺-NTA or glutathione (GSH) agarose beads before GST/His pull-down assays. GST-tagged EZH2 and His-tagged UL44 were negative controls. **(C)** Co-localization of FHL2 with UL84 in HFF

cells. pcDNA3.1-UL84 or pcDNA3.1-FHL2 were transiently transfected in HFFs for 48 h, cells were fixed and permeabilized, followed by incubation with anti-FHL2/anti-UL84 antibody primary Abs and with Dylight 488/647-conjugated fluorescent secondary Abs before confocal microscopy. Sub-cellular localization of FHL2 (Green), UL84 (Red), DAPI (Blue) were examined. **(D)** Endogenous Co-IP analysis between UL84 and FHL2 under HCMV infection. HFF cells were infected with HCMV (MOI = 1) for 24 h, followed by Co-IP and immunoblot analysis using anti-UL84, anti-EZH2 or anti-FHL2 antibody as the indicated respectively. **(E)** Schematic diagrams of Flag-tagged FHL2 and its truncated proteins are shown, with the LIM domains highlighted as red boxes. **(F)** Determination of the interaction between UL84 and truncates of FHL2. Full-length Myc-UL84 and truncated plasmids of Flag-FHL2 were transiently co-transfected into HEK293T cells respectively for 48 h before Co-IP and immunoblot analysis. **(G)** Schematic diagrams of Myc-tagged UL84 and its truncated proteins, along with functional analysis of UL84 structural domains, are shown below. **(H)** Determination of the interaction between FHL2 and truncates of UL84. Full-length Flag-FHL2 and truncated plasmids of Myc-UL84 were transiently co-transfected into HEK293T cells respectively for 48 h before Co-IP and immunoblot analysis. **(I)** Molecular docking simulation of the complex of UL84 and FHL2. The structure of the FHL2 and UL84 was downloaded from the Protein Data Bank, green refers to FHL2, blue refers to UL84. **(J)** Con-RNAi, UL84-RNAi-#1, UL84-RNAi-#1 with UL84-mut, UL84-RNAi-#1 with UL84(Δ400-460) were transiently transfected in HFFs for 24 h, respectively. After transfection, HFFs were infected with HCMV-WT. Supernatants were harvested at 48, 60, 72, and 96 hpi for measurement of viral titers. **(K)** Con-RNAi, UL84-RNAi-#1, UL84-RNAi-#1 with UL84-mut, UL84-RNAi-#1 with UL84(Δ400-460) were transiently transfected in HFFs for 24 h respectively, then infected with HCMV (MOI = 1). Samples were harvested at different time points. One portion of the samples was analyzed for *IFNB1*, normalized against GAPDH mRNA levels before RT-qPCR analysis. **(L)** Con-RNAi, UL84-RNAi-#1, UL84-RNAi-#1 with UL84-mut, UL84-RNAi-#1 with UL84(Δ400-460) were transiently transfected in HFFs. After transfection, HFFs were infected with HCMV-WT. Supernatants were harvested at 48, 60, 72, and 96 hpi for measurement of viral titers. All experimental assays were conducted in triplicate with independent biological replicates. For all figures, statistical analyses were performed using two-tailed t-test. Differences were considered statistically significant when * denoted $p < 0.05$, ** denoted $p < 0.01$, *** denoted $p < 0.001$, and **** denoted $p < 0.0001$.

Subsequently, we investigated whether the regulatory effects of UL84 on interferon-β and the impact on viral titer in the context of HCMV infection were dependent on its interaction with FHL2. However, as previously reported, UL84 is an essential gene for HCMV replication, which renders the construction of UL84-deficient virus strains infeasible [39]. We also attempted to generate one and complement its function using UL84-expressing cells, but without success. Therefore, we constructed an expression plasmid containing UL84 mutations by introducing synonymous mutations into the region of the UL84-RNAi-#1 target sequence (UL84-mut), which preserved the UL84 amino acid sequence while conferring resistance to UL84-RNAi-#1 (Fig 2J). As shown in Fig 2K, UL84 synonymous mutants (UL84-mut) can still significantly repress mRNA levels of IFN-β induced by HCMV infection in the presence of UL84-RNAi-#1. In contrast, UL84 (Δ400–460) mutants, which lack the interaction region with FHL2, fail to suppress IFN-β expression when co-expressed with UL84-RNAi-#1. Meanwhile, viral titer assays confirmed that the C-terminal domain of UL84 (400–460 aa) was also required for efficient viral replication (Fig 2L). Collectively, these data imply the important role of the interaction between UL84 and FHL2 in subverting host immune defenses and supporting virus reproduction.

## Virus infection triggers the phosphorylation and nuclear translocation of FHL2

It is noteworthy that the abundance of FHL2 in HFF cells exhibits but a transient increase following HCMV infection (Fig 3A). Furthermore, HFF cells infected with IAV, EV71, HSV and ZIKV also demonstrate an immediate upregulation of FHL2 expression (Fig 3B–3E). Next, we examined the distribution of FHL2 in uninfected HFF cells and assessed whether viral infection had any impact on its localization. Indirect immunofluorescence assays (IFA) showed that FHL2 was mainly distributed in the cytoplasm of uninfected cells (Fig 3F). However, after 6 hours of HCMV infection, FHL2 was predominantly translocated into the nucleus. Similarly, we observed that IAV, EV71, HSV, and ZIKV also exhibit the phenomenon of infection-induced translocation of FHL2 into the nucleus (Fig 3F).

Phosphorylation has been widely reported to activate many cellular activators or co-activators, leading to their translocation into the nucleus and subsequent induction of IFN-I transcription [40]. Research indicates that following the entry of the virus into the cell, IRF3 undergoes phosphorylation and is translocated to the nucleus, where it binds to the CBP/p300 co-activator complex and activates the transcription of the *IFNB1* gene [7]. However, the molecular mechanisms underlying FHL2 nuclear translocation remain incompletely elucidated. Notably, domain analysis failed to identify the classical nuclear localization signal (NLS) or nuclear export signal (NES) motifs, implying that its nuclear transport may

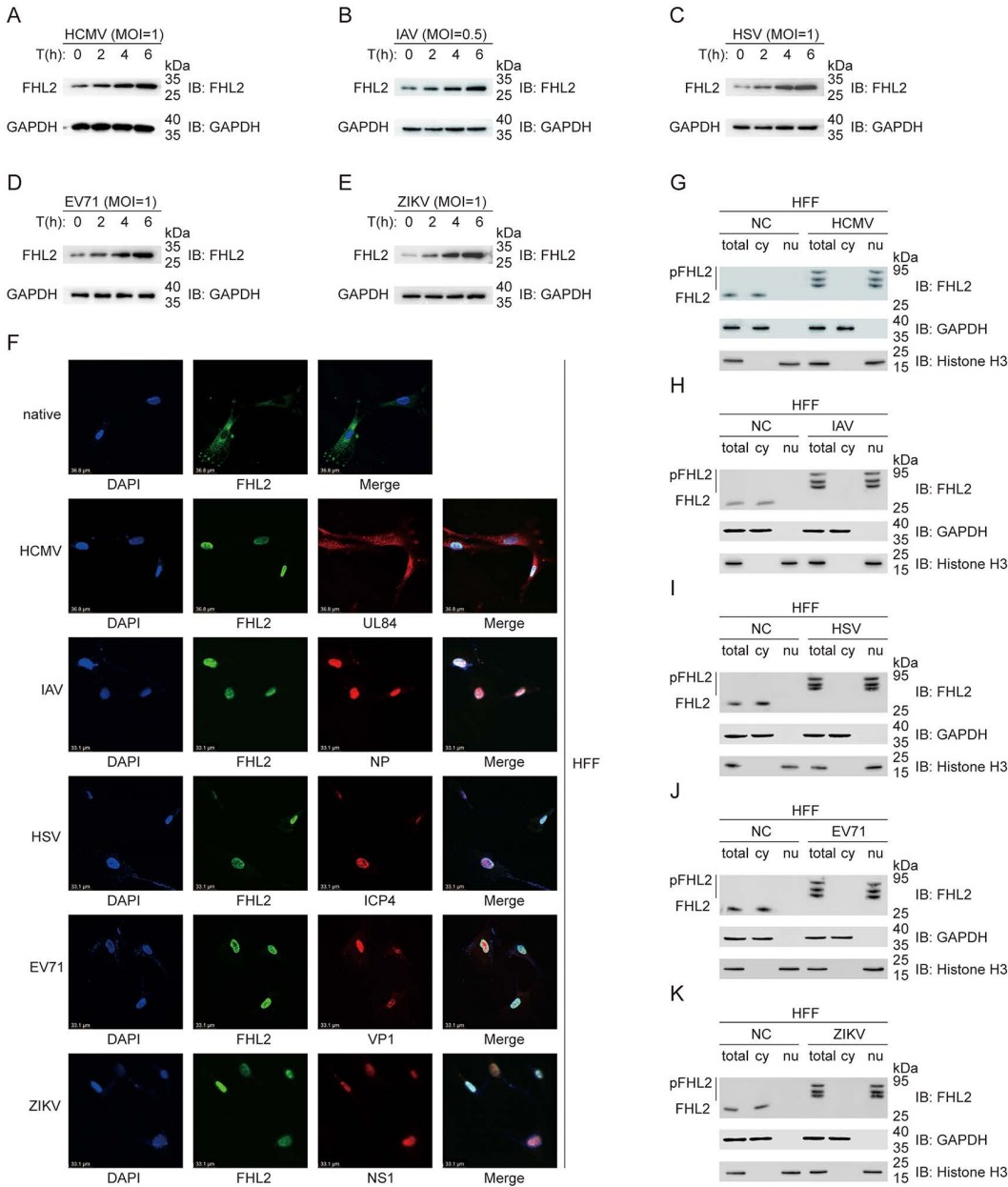

**Fig 3. Viruses triggered upregulation and phosphorylation of FHL2. (A)** Immunoblot analysis of FHL2 abundance under HCMV infection. HFF cells were infected with HCMV (MOI = 1) for 2, 4 and 6 h, and whole cell lysates were prepared and detected with Western blot. **(B–E)** Immunoblot analysis of FHL2 abundance under IAV, HSV, EV71, or ZIKV infection. HFF cells were infected with IAV (MOI = 0.5), HSV (MOI = 1), EV71 (MOI = 1), or ZIKV (MOI = 1) respectively for 2, 4 and 6 h, and whole cell lysates were prepared and detected with Western blot. **(F)** Nuclear localization of endogenous FHL2 under HCMV, IAV, HSV, EV71, or ZIKV infection. HCMV (MOI = 1), IAV (MOI = 0.5), HSV (MOI = 1), EV71 (MOI = 1), or ZIKV (MOI = 1) infected HFFs for 6 h, cells were fixed and permeabilized, followed by incubation with anti-FHL2 primary Abs and then with Dylight 647-conjugated fluorescent secondary Abs before confocal microscopy. Subcellular localization of FHL2 (Green), representative protein expressed by indicated virus (Red), DAPI (Blue) were observed by confocal microscopy. **(G)** Phosphorylation of FHL2 under HCMV infection by Phos-tag and immunoblot analysis. HCMV (MOI = 1) infected HFFs for 6 h, cytoplasmic and nuclear fractions were prepared using a Nuclear and Cytoplasmic Protein Extraction Kit before Phos-tag SDS-PAGE and immunoblot analysis. **(H–K)** Phosphorylation of FHL2 under IAV, HSV, EV71, or ZIKV infection by Phos-tag and immunoblot analysis. IAV (MOI = 0.5), HSV (MOI = 1), EV71 (MOI = 1), or ZIKV (MOI = 1) respectively infected HFF cells for 6 h, cytoplasmic and nuclear fractions were prepared using a Nuclear and Cytoplasmic Protein Extraction Kit before Phos-tag SDS-PAGE and immunoblot analysis. The displayed images were representative ones from three independent experiments.

be regulated by post-translational modification-mediated processes. Therefore, we conducted nuclear and cytoplasmic extraction and phosphate-affinity SDS-PAGE analysis to investigate whether the FHL2 nuclear translocation is triggered by virus-induced phosphorylation of FHL2. Phos-tag gel electrophoresis and Western blot analysis further confirmed that FHL2 was predominantly localized in the cytoplasm without viral induction, and remained unphosphorylated (Fig 3G, Lanes 1 and 2). However, upon viral infection, FHL2 underwent phosphorylation and translocated into the nucleus (Fig 3G, Lane 6). This phenomenon was also observed in HFF cells infected with IAV, EV71, HSV and ZIKV (Fig 3H–3K). These results indicate that virus infection induces the phosphorylation and nuclear translocation of FHL2, which may serve as part of a signaling cascade for detecting viral invasion.

## FHL2 enhances the virus-triggered IFN-β response

It has been reported that FHL2 can interact with ATF2/c-Jun and CBP/p300, which are general constituents of the IFN-β transcriptional enhanceosome [41,42]. Therefore, we proceeded to investigate the potential involvement of FHL2 in IFN-I transcription. As expected, exogenously expressed FHL2 could enhance the activation of IFN-β promoter induced by HCMV in a dose-dependent manner and time-dependent manner (Fig 4A and 4B). In addition, FHL2 was found to enhance the transcription of IFN-β promoter induced by various viruses, such as IAV, EV71, HSV and ZIKV in a similar manner (S5A–S5H Fig). In contrast, interferon-β promoter activity was significantly inhibited in FHL2-knockout cell lines (Fig 4C). Moreover, upon HCMV stimulation of FHL2-overexpressing cells, the transcriptional mRNA levels of *IFN-B1* and downstream *ISG15* and *IFIT1* were significantly elevated compared to vector-transfected controls (Fig 4D). Whereas the expression of downstream interferon-stimulated genes was significantly decreased in FHL2-knockout cell lines, as indicated by RT-qPCR analysis (Fig 4E). This phenomenon was also observed in HFF cells infected with IAV, EV71, HSV, and ZIKV (S5I–S5L Fig).

Both the upregulation of mRNA levels for interferon-stimulated genes (*ISGs*) and phosphorylation of STAT1 are critical indicators for assessing type I interferon activation [43,44]. We also observed a significant increase in STAT1 phosphorylation in FHL2-overexpressing cell line following HCMV infection (Fig 4F). Meanwhile, in the FHL2-knockout cell line, the STAT1 phosphorylation was significantly inhibited compared with the control group (Fig 4G). Given that the phosphorylation of IRF3 represents the final step in the signaling pathway of the type I interferon response, we also assessed FHL2's role in this signaling cascade by quantifying IRF3 phosphorylation dynamics in FHL2-overexpressing and FHL2-knockout cell lines [7,45]. The results indicated that neither overexpression nor knockout of FHL2 had an impact on the phosphorylation of IRF3 (Fig 4F and 4G). These results indicate FHL2 may enhance IFN-β transcriptional activation rather than the cascade signaling activation of viral infection.

## FHL2 facilitates formation of IFN-β transcriptional enhanceosome complex

**FHL2 bridges TBP.** Next, we prompted to determine the mechanism for how FHL2 regulates IFN-I signaling. Given that FHL2 lacks nucleic acid binding capability and cannot directly recognize DNA, clarifying its mechanism for participating in interferon transcriptional activation requires an investigation of its relationship with both upstream cis-acting regulatory elements and the enhanceosome complex within the interferon promoter [18] The schematic in Fig 5A depicts *IFNB1* promoter region (-499 to +100) of the *Homo sapiens*, sourced from EPD (https://epd.expasy.org/epd/). It annotates key transcriptional factor binding motifs, including the IRF3 consensus sequence 5'-AANNGAAA-3' recognized by IRF3 [46], consensus sequence 5'-TGACATAG-3' recognized by c-Jun [47], and consensus sequence 5'-GGGRNNYCC-3' recognized by NF-κB [48]. To achieve this, we first designed three sets of specific q-PCR probes targeting distinct regions within the IFN-β promoter. Through Chromatin Immunoprecipitation (ChIP) assay using FHL2-specific antibody, we observed that FHL2 was predominantly enriched in the P2 region of the IFN-β promoter, indicating its direct involvement in transcriptional regulation of IFN-β (Fig 5B).

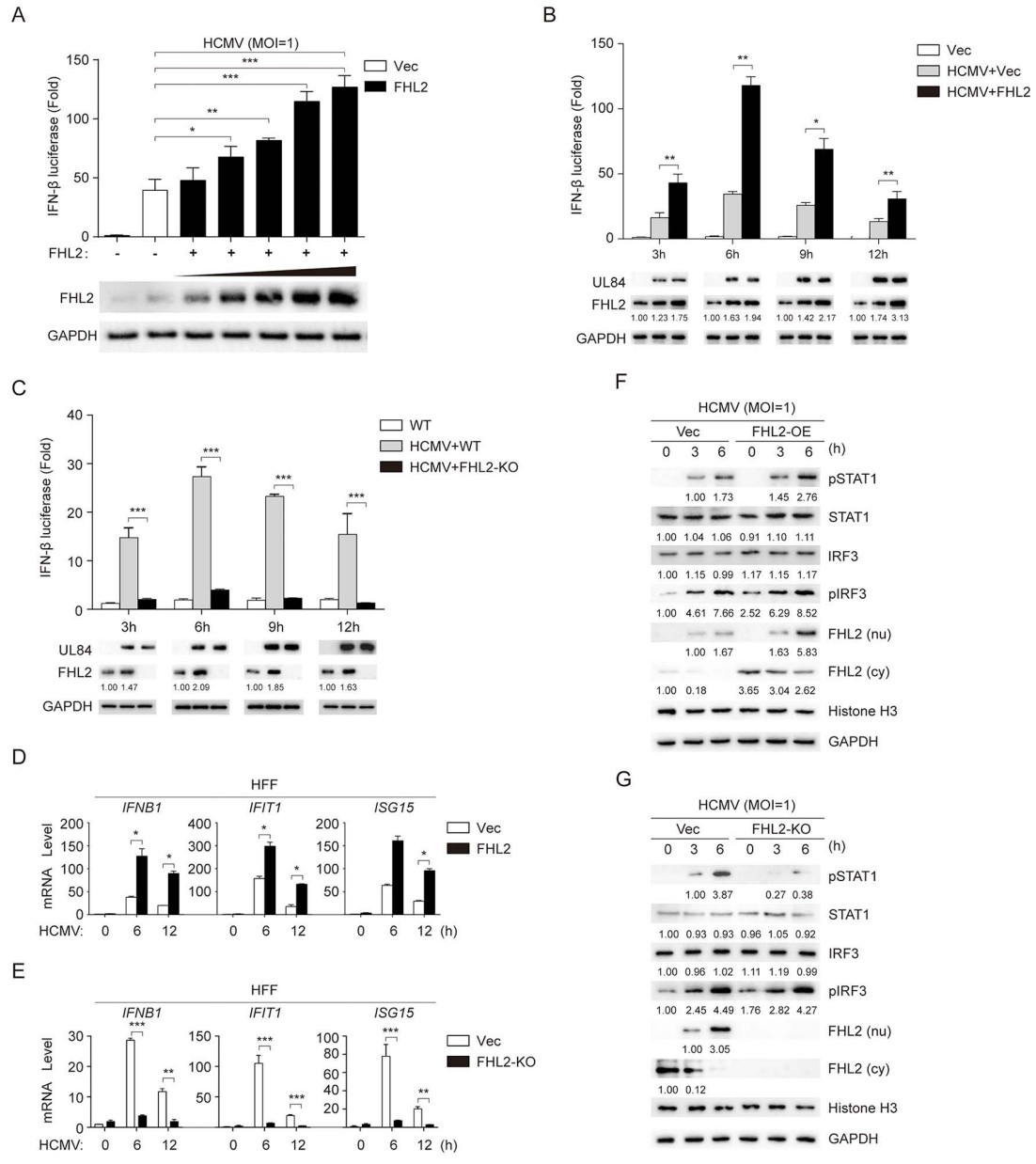

**Fig 4. FHL2 enhances HCMV-triggered IFN-β promoter activity and downstream interferon-stimulated gene responses. (A)** HFF cells were transfected with pGL3-luci-IFN-beta and pRL-TK reporter plasmids, along with increasing amount of the pcDNA3.1-FHL2 expression vector or the pcDNA3.1 empty vector as a control for 24 h, the cells were infected with HCMV (MOI = 1) for 24 h before the dual luciferase reporter assay. **(B)** HFF cells were transfected with the plasmid of pGL3-luci-IFN-beta and pRL-TK together with pcDNA3.1-FHL2 or the empty vector for 24 h, and then infected with HCMV (MOI = 1) for the indicated times before conducting dual-luciferase reporter assay. Another portion was subjected to Western blotting. (C) FHL2-knockout cell line was transfected with the plasmid of pGL3-luci-IFN-beta and pRL-TK for 24 h, followed by a 24 h, and then infected with HCMV (MOI = 1) for the indicated times before conducting dual-luciferase reporter assay. Another portion was subjected to Western blotting. **(D)** HFF cells were transfected with the plasmid of pcDNA3.1-FHL2 or the empty vector for 24 h, and then infected with HCMV (MOI = 1) for the indicated times before RT-qPCR analysis of the indicated antiviral interferon-stimulated genes. **(E)** FHL2-knockout cell line was infected with HCMV (MOI = 1) for the indicated times before RT-qPCR analysis of the indicated antiviral interferon-stimulated genes. (F) The FHL2-overexpressing cell line was infected with HCMV (MOI = 1), followed by Western blotting analysis of pSTAT1, STAT1, cytoplasmic FHL2 (cy), and nuclear FHL2 (nu). **(G)** The FHL2-knockout cell line was infected with HCMV (MOI = 1), followed by Western blotting analysis of pSTAT1, STAT1, cytoplasmic FHL2 (cy), and nuclear FHL2 (nu). The displayed images were representative ones from three independent experiments. For all figures, statistical analyses were performed using two-tailed t-test. Differences were considered statistically significant when * denoted p < 0.05, ** denoted p < 0.01, *** denoted p < 0.001, and **** denoted p < 0.0001.

According to previous reports, the formation of IFN-β transcriptional enhanceosome is a complicated regulatory process that involves core backbone components such as general transcription factors ATF-2/c-Jun and CBP/p300, as well as IFN-β promoter specific factor IRF3, followed by the recruitment of HATs, GCN5, and other multifaceted regulators [12,49]. Moreover, FHL2 has been reported to associate with c-Jun (1–100 aa) and p300 (1–670 aa) to form complexes, which synergistically stimulate the transactivating activities of AP-1 and β-catenin [21,42]. ChIP assay showed that IRF3 could be co-immunoprecipitated with FHL2 while specifically binding to IFN-β promoter under HCMV infection. Meanwhile, FHL2 could also specifically bind to the IFN-β promoter and be co-immunoprecipitated with IRF3, c-Jun, and p300 as part of complex formations (Fig 5C). Furthermore, we conducted Co-IP assays to further analyze the interaction between FHL2 and c-Jun as well as p300, especially IRF3, aiming to elucidate whether FHL2 interacts with these transcription factors to form complexes that promote IFN-β transcription. The results showed that FHL2 not only interacted with the general transcription factors c-Jun and p300 as previously reported, but also interacted with the promoter-specific transcription factor IRF3 upon ectopic co-expression of plasmids (Fig 5D). However, through Co-IP analysis, we could observe that FHL2 would lose its interaction with IRF3, c-Jun and p300 when the LIM2 domain of FHL2 was absent, indicating that this domain may play a key role in the formation of IFN-β transcriptional enhanceosome complex (Fig 5E).

It has been reported the phosphorylation of the Y93 residue in FHL2 by FAK is a critical factor for nuclear entry [50]. Generally, phosphorylation modification plays a crucial role in the nuclear translocation and functional activation of interferon regulators [40]. A typical example is IRF3, whose phosphorylation site is located in the form of a C-terminal phospho-acceptor cluster, which is essential not only for nuclear translocation but also for dimerization [51]. Therefore, we hypothesize that phosphorylation clusters may exist around the amino acid residue 93. Based on the NetPhos-3.1-Services-DTU Health Tech analysis, we have predicted a potential phospho-acceptor cluster within the LIM2 domain, which includes possible phosphorylation motifs at the Y97, S98, S99, and T112 amino acid sites. As illustrated in Fig 5F, mutations at any of these four sites would abolish the interaction between FHL2 and IRF3, c-Jun, or p300. Additionally, to rule out the possibility that some or all of these sites are crucial for the correct folding of the FHL2 protein, we carried out molecular dynamics simulations between the wild-type and mutant proteins. Initially, we undertook structural preparation and modeling of the FHL2 protein (Fig 5G). The molecular dynamics simulations revealed the stability of both the wild-type and mutant forms of FHL2, as evidenced by Radius of gyration (Rg), root means square deviation (RMSD) and root means square fluctuation (RMSF) trajectories (Fig 5H–5J). These findings are further corroborated by a SASA plot, indicating that mutations at these four critical amino acid sites do not affect the protein's structural characteristics or stability (Fig 5K).

Collectively, these findings indicate that all four predicted sites (Y97, S98, S99, and T112) within the LIM2 domain are essential for FHL2's interaction with IRF3, c-Jun, or p300 during the assembly of the IFN-β enhanceosome complex.

**FHL2 bridges TBP recruitment to promote IFN-β transcriptional response.** We have demonstrated that the four residue sites (Y97, S98, S99, and T112) within the LIM2 domain are critical for the formation of IFN-β enhanceosome complex. Generally, the enhanceosome complex subsequently recruits TFIID to the TATA-box of the promoter, triggering the transcription of downstream genes [40]. TFIID is a multi-subunit complex consisting of TATA-binding protein (TBP) and TBP-associated factors (TAF) [52,53], and the function of the general transcription factors TBP is to bind the TATA-box and bend DNA [53]. Indeed, ChIP assay under HCMV infection showed that IRF3 could be co-immunoprecipitated with TBP while specifically binding to IFN-β promoter, while TBP could also be co-immunoprecipitated with FHL2 as part of enhanceosome complex formations (Fig 6A). However, Co-IP assays in HEK293T cells co-transfected with TBP and plasmids expressing IRF3, p300, and c-Jun showed that none of these proteins were directly co-precipitated with TBP, either individually or together (S7 Fig). Meanwhile, when we knocked out the endogenous FHL2, TBP levels are decreased in IRF3 enrichment, although it still recruited p300 and c-Jun to the IFN-β promoter (Fig 6B). Molecular docking simulation of the IRF3-FHL2-TBP complex indicates that IRF3 specifically recognizes the N-terminal LIM domain of FHL2, while TBP associates with the C-terminal region of FHL2 (Figs 6C and S6E). Therefore, we speculated that the mechanism by which FHL2 regulates IFN-β transcription may involve the recruitment of TFIID to the TATA-box.

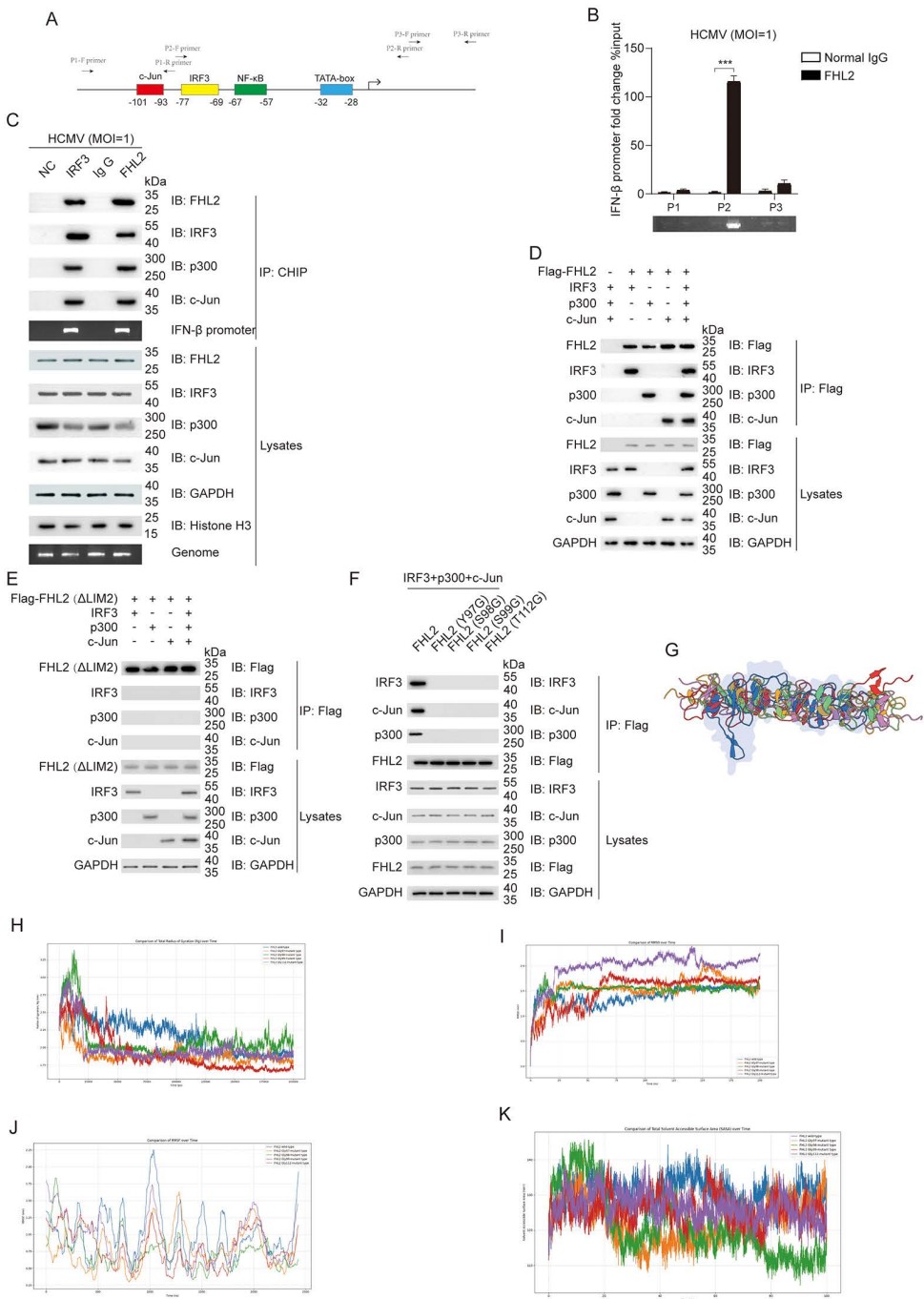

**Fig 5. Interactions between FHL2 and components of IFN-β enhanceosome complex. (A)** Schematic diagram of the enhancer region within the *IFNB1* promoter. **(B)** The involvement of FHL2 in the regulation of IFN-β transcription was identified through the ChIP assay. HFF cells were infected with HCMV (MOI = 1) for 6 h, followed by ChIP assays using FHL2-specific or H3 histone control monoclonal antibodies, and IFN-β enhancer-specific P1, P2, and P3 primers. **(C)** FHL2 associated with IFN-β enhanceosome complex. HFFs were infected with HCMV (MOI = 1) for 6h, and cross-linked with 1% formaldehyde for 15min before CHIP. NC as non-specific control, normal rabbit IgG as a negative control, anti-IRF3 positive control, anti-FHL2 as an inductive antibody. **(D)** Co-IP assays were performed to investigate the interaction between FHL2 and IRF3, c-Jun, or p300. HEK293T cells were transiently co-transfected with pcDNA3.1-FHL2, pcDNA3.1-IRF3, pcDNA3.1-p300, and pcDNA3.1-c-Jun plasmids for 48 h. Total cell lysates were then prepared and immunoprecipitated with FHL2-specific antibody before being analyzed by Western blot using FHL2, IRF3, p300, and c-Jun antibodies. **(E)** Co-IP assay of FHL2 (ΔLIM2) with IRF3, c-Jun, or p300. Total cell lysates were prepared from HEK293T cells co-transfected with pRK-Flag-FHL2

(ΔLIM2), pcDNA3.1-IRF3, pcDNA3.1-p300, and pcDNA3.1-c-Jun plasmids for 48 h and then immunoprecipitated with FHL2-specific antibody before Western blot analysis. **(F)** Effect of different mutations on the interaction between FHL2 and c-Jun, IRF3, or p300. HEK293T cells were transiently co-transfected with pRK-Flag-FHL2 (full length or four single point mutants) and pcDNA3.1-c-Jun, pcDNA3.1-IRF3 or pcDNA3.1-p300 respectively for 48 h before Co-IP analysis. The displayed images were representative ones from three independent experiments. **(G)** Schematic diagram of the overall structural conformation of FHL2. **(H)** Figure of the radius of gyration results, a core indicator for characterizing molecular spatial conformation in molecular dynamics simulation. **(I)** Figure of the root mean square deviation results, a core indicator for characterizing molecular spatial conformation in molecular dynamics simulation. **(J)** Figure of the root-mean-square fluctuation results, a core indicator for characterizing molecular spatial conformation in molecular dynamics simulation. **(K)** Figure of the Solvent accessible surface area results, a core indicator for characterizing molecular spatial conformation in molecular dynamics simulation. The displayed images were representative ones from three independent experiments. For all figures, statistical analyses were performed using two-tailed t-test. Differences were considered statistically significant when * denoted $p < 0.05$, ** denoted $p < 0.01$, *** denoted $p < 0.001$, and **** denoted $p < 0.0001$.

Next, when FHL2 was introduced into the co-expressed cells, it was found that FHL2 could be co-immunoprecipitated with both TBP and the typical components of the IFN-β transcriptional enhanceosome complex (Fig 6D). However, when IRF3, p300, or c-Jun were co-transfected with TBP and the FHL2 LIM2 deletion mutant (ΔLIM2), the Co-IP results indicated that only TBP was able to bind to ΔLIM2, while IRF3, p300, and c-Jun were not detected in association with ΔLIM2 during co-immunoprecipitation (Fig 6E). Moreover, when we analyzed the interaction of TBP with FHL2 $^{Y97G}$, FHL2 $^{S98G}$, FHL2 $^{S99G}$, or FHL2 $^{T112G}$ in HEK293T cells, the results showed that TBP protein was able to interact with all four point-mutated FHL2 proteins, indicating the binding site of TBP did not overlap with the complex of IRF3, p300, and c-Jun at the LIM2 domain of FHL2 (Fig 6F). Furthermore, molecular docking of the complex formation of IRF3-FHL2-TBP demonstrates that IRF3 specifically recognizes the N-terminal LIM domain of FHL2, while TBP binds to its C-terminal region (Figs 6C and S6E). Together, these findings suggest that the LIM2 domain of FHL2 is not a dominant binding site for TBP and that TBP does not compete for binding at sites occupied by typical components of the IFN-β transcription enhancer complex. Therefore, we assumed that FHL2 may function as a bridge connecting the enhanceosome to the TFIID complex, providing an interface for protein-protein interaction (PPI).

## UL84 disrupts formation of the IRF3-c-Jun-p300 complex by interacting with FHL2

HCMV UL84 has been previously reported to be a replication essential protein [54]. From the above, we have demonstrated that UL84 inhibits the host antiviral response. Here, we further investigate how the interplay between UL84 and FHL2 contributes to the viral immune evasion from innate immunity.

Firstly, we further verified in an *in vitro* system through luciferase reporter gene assays that the inhibitory effect of UL84 on HCMV-induced IFN-β depends on its interaction with FHL2. As shown in Fig 7A, compared with wild-type, UL84 (Δ400–460) mutants, which lack the interaction region with FHL2, fail to suppress the dose-dependent enhancement of IFN-β promoter activity by FHL2. Secondly, in UL84 overexpression and specific siRNA-knockdown cell lines, by detecting changes in the phosphorylation of STAT1 and IRF3, we confirmed that UL84 may inhibit IFN-β transcriptional activation rather than the cascade signaling activation induced by viral infection (Fig 7B–7D).

We have shown that LIM2 domain of FHL2 is essential for virus-triggered IFN-β response. In addition, LIM2 domain is a key structure for UL84 to interact with FHL2. Therefore, it is necessary to investigate the interplay between HCMV UL84 and the IRF3-c-Jun-p300 complex. Our results indicated that UL84 but not its truncation mutant (Δ400–460) abolished the interactions between FHL2 and IRF3, c-Jun or p300, disrupting the formation of IFN-β enhanceosome complex (Fig 7E–7G). Moreover, we have determined that the amino acid residues Y97, S98, S99 and T112 within FHL2 are important for IRF3-c-Jun-p300 reciprocal recognition, so we are wondering if these sites are also essential for UL84 binding? To investigate this, we transient co-transfected UL84 with FHL2$^{Y97G}$, FHL2$^{S98G}$, FHL2$^{S99G}$, or FHL2$^{T112G}$ in HEK293T cells, respectively. The results showed that UL84 protein was not able to interact with FHL2$^{T112G}$ (Fig 7H), indicating that UL84 competitively binds to FHL2 through T112 site, and therefore inhibits the interplay between FHL2 and the IRF3-c-Jun-p300 complex.

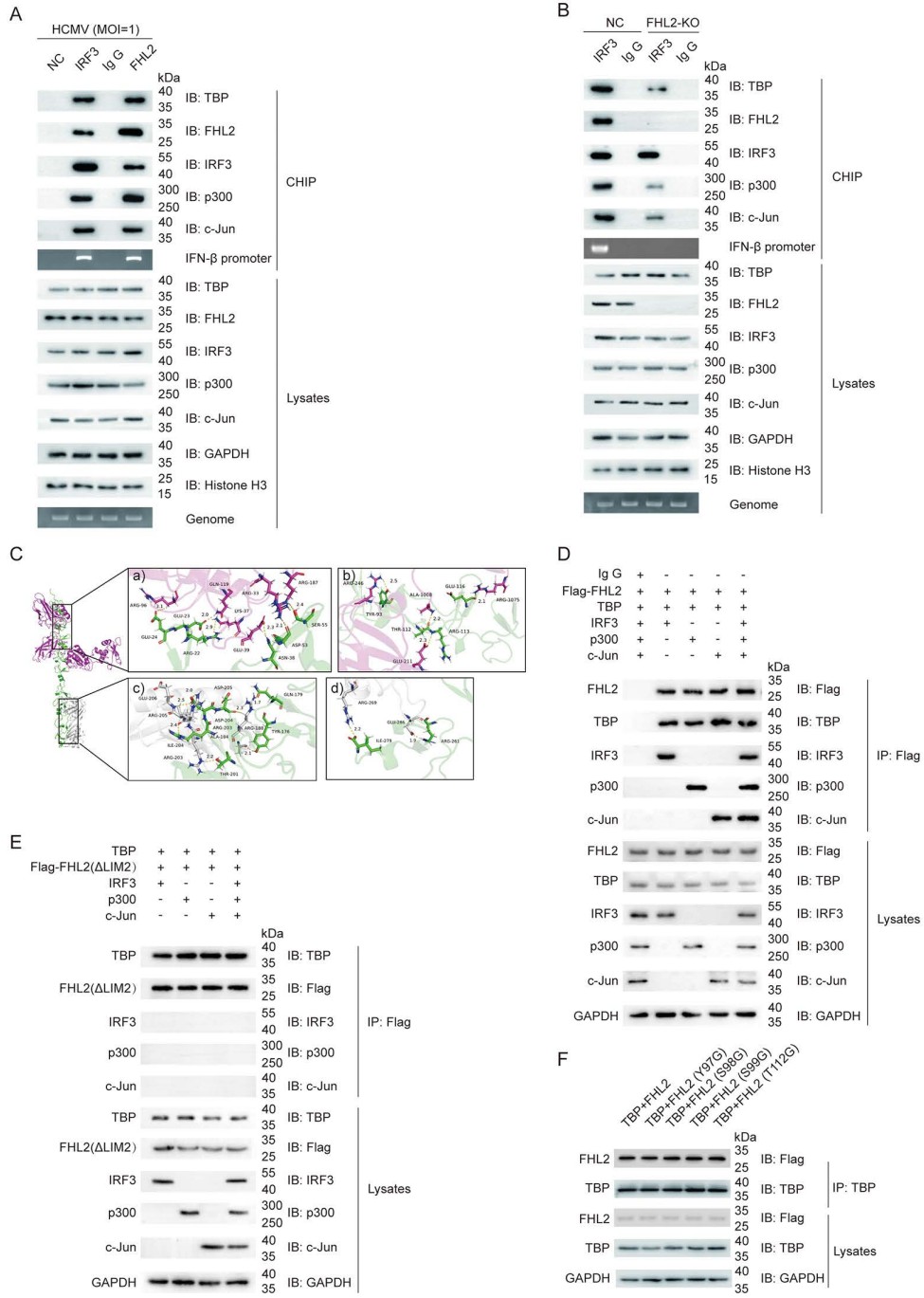

**Fig 6. FHL2 promotes the recruitment of TBP to PIC. (A)** FHL2 associated with IFN-β PIC complex. HFFs were infected with HCMV (MOI = 1) for 6h, and cross-linked with 1% formaldehyde for 15min before CHIP. NC as non-specific control, normal rabbit IgG as a negative control, anti-IRF3 positive control, anti-FHL2 as an inductive antibody. **(B)** FHL2 associated with IFN-β PIC complex. HFFs and FHL2-knockot cell line were infected with HCMV (MOI = 1) for 6h, and cross-linked with 1% formaldehyde for 15min before CHIP. Normal rabbit IgG as a negative control, anti-IRF3 positive control. **(C)** Molecular docking simulation of the complex of IRF3-FHL2-TBP. The structure of the IRF3, FHL2 and TBP was downloaded from the Protein Data Bank, green refers to FHL2, grey refers to TBP, and pink refers to IRF3. **(D)** Co-IP analysis of FHL2 interactions with TBP, IRF3, c-Jun, or p300. Expression plasmids of Flag-FHL2, TBP, IRF3, p300 and c-Jun were transiently transfected into HEK293T cells for 48h, and then the total cell lysates were prepared and immunoprecipitated with anti-Flag, anti-TBP, anti-IRF3, anti-c-Jun or anti-p300 antibodies, followed by immunoblot analysis. **(E)** Co-IP analysis of FHL2 (ΔLIM2) interactions with TBP, IRF3, c-Jun, or p300. Expression plasmids of Flag-FHL2 (ΔLIM2), TBP, IRF3, p300 and c-Jun were

transiently transfected into HEK293T cells for 48 h, and then the total cell lysates were prepared and immunoprecipitated with anti-Flag, anti-TBP, anti-IRF3, anti-c-Jun or anti-p300 antibodies, followed by immunoblot analysis. **(F)** Effect of different mutations on the interaction of FHL2 with TBP. HEK293T cells were transiently transfected with Flag-tagged FHL2 (full-length or the four single-point mutants) and pcDNA3.1-TBP for 48 h before Co-IP analysis was performed. The displayed images were representative ones from three independent experiments.

Considering FHL2 is proven to be a key regulator in the mechanism of IFN-β transcription, we speculated that the mechanism of HCMV evasion of innate immunity might be: UL84 entries the nucleus of host cell to hijack TBP, blocking its recruitment to the TATA-box of IFN-β. To verify our conjecture, plasmids encoding UL84, TBP were transiently co-transfected with FHL2 or FHL2$^{T112G}$ into HEK293T cells. Co-IP assays showed that UL84, as a prey protein, was co-immunoprecipitated with TBP in the presence of FHL2 (Fig 7I, lane 1). In contrast, when co-transfected with FHL2$^{T112G}$, UL84 could not enrich TBP, indicating an indirect protein interaction between UL84 and TBP (Fig 7I, lane 2). Meanwhile, considering LIM2 is the structural domain recognized by UL84 interacting with FHL2, and TBP interacts with FHL2 does not depend on LIM2. Therefore, when HCMV UL84 enters the nucleus to regulate innate immunity, it occupies the LIM2 domain of FHL2, while TBP possibly occupies the LIM3 or LIM4 structural domain, and thereby hijacking TBP from being recruited to the TATA-box of IFN-β (Figs 7J and S6F).

### Phosphorylation of the amino acid residue T112 is critical for FHL2-mediated antiviral response

Because the amino acid residue T112 of FHL2 is essential for both the IRF3-c-Jun-p300 assembling and for UL84 to interact with FHL2, we next examined the role of T112 in the type I interferon responses. The luciferase reporter assay results showed that overexpression of FHL$^{T112G}$ (as a dominant-negative control) repressed the activation of the IFN-β promoter in a dose-dependent manner (Fig 8A). Meanwhile, RT-qPCR analysis revealed that FHL2 (T112G) not only failed to enhance the expression of *IFNB1* induced by HCMV infection, but instead exerted a certain inhibitory effect (Fig 8B). In addition, the phosphate affinity immunoblot analysis showed that there were two major phosphorylation bands of FHL2$^{T112G}$ appeared during HCMV infection (Fig 8C), which was one band less than that of the wild-type FHL2 (Fig 3G–3K). These results suggest that T112 is one of the amino acids residues that be phosphorylated. Previous studies suggest that Y93, Y97, Y176, Y217, Y236, and S238 of FHL2 can be phosphorylated upon certain conditions [50,55,56], next we investigated the roles of Y97, S98, S99, and T112 of FHL2 in virus-triggered FHL2 nuclear translocation. The results indicated that almost all FHL2$^{T112G}$ was induced into the nucleus triggered by HCMV infection in HFFs (Fig 8D). In contrast, only half of FHL2 $^{Y97G, S98G, S99G, T112G}$ was induced into the nucleus (Fig 8D). Also, confocal microscopy showed that half of FHL2(ΔLIM2) was induced into the nucleus triggered by HCMV infection in HFFs (Fig 8E). These results suggest that Y97G, S98G, S99G, and T112G of FHL2 are essential for virus-triggered of IFN-β transcriptional response. However, phosphorylation of currently unknown sites, triggered by HCMV infection, could also cause the translocation of FHL2 into the nucleus.

Since we have found that T112 of FHL2 is phosphorylated by virus infection, we next tried to identify the kinase which triggers the phosphorylation of FHL2 thus drives it into the nucleus. Focal adhesion kinase (FAK) is known to phosphorylate the tyrosine of FHL2 to affect cell proliferation, but not related to host innate immunity [50]. According to NetPhos-3.1-Services-DTU Health Tech analysis, PKC may play an important influence on T112 phosphorylation. Therefore, we selected the PKC inhibitor LY333531 and specific targeted siRNA to investigate whether it has any effect on FHL2 nuclear translocation (Fig 8F–8H). The results showed that the amount of FHL2 translocated into nucleus was significantly reduced both in the inhibitor-treated and siRNA-knockdown groups after HCMV infection. These results suggest that PKC may be involved in phosphorylating FHL2. However, nuclear translocation of FHL2 was not fully inhibited, suggesting other yet-unknown phosphokinases may be involved.

Therefore, the phosphorylation of FHL2 at the T112 site catalyzed by PKC is crucial for FHL2 rapid translocation to the nucleus, and participation in immune response regulation.

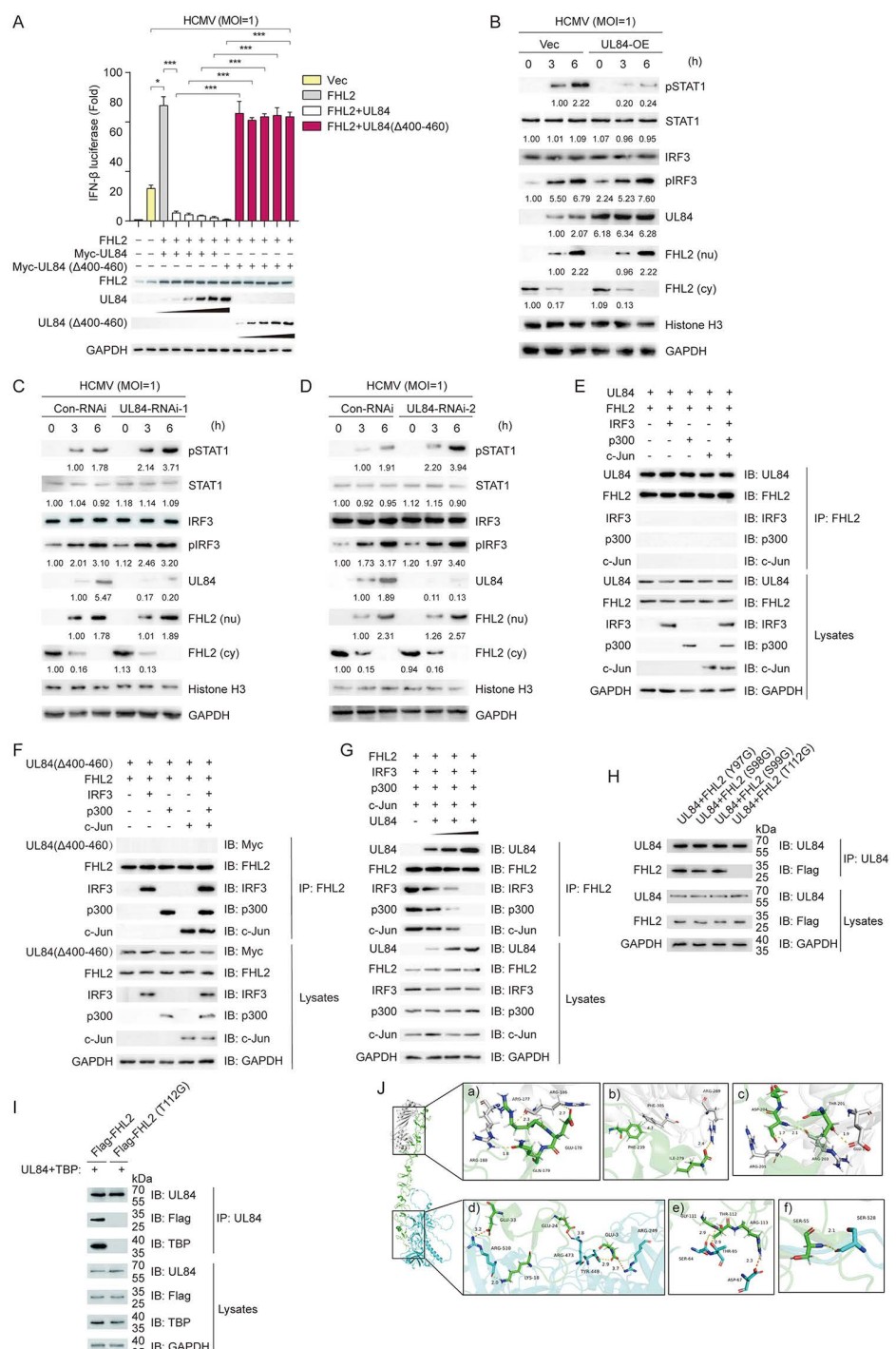

**Fig 7. UL84 inhibits the transcription of IFN-β. (A)** UL84 interacts with FHL2 to repress IFN-β transcription. HFFs were transiently transfected with expression plasmids for pGL3-luci-IFN-beta, FHL2, UL84, UL84 (Δ400-460) for 24 h, HCMV (MOI = 1) reinfection of cells for 6h before luciferase assay. **(B)** The UL84-overexpressing cell line was infected with HCMV (MOI = 1), followed by Western blotting analysis of pSTAT1, STAT1, IRF3, pIRF3, cytoplasmic FHL2 (cy), and nuclear FHL2 (nu). **(C)** HFF cells were transfected with the Con-RNAi, UL84-RNAi-#1, and then infected with HCMV (MOI = 1), followed by Western blotting analysis of pSTAT1, STAT1, IRF3, pIRF3, cytoplasmic FHL2 (cy), and nuclear FHL2 (nu). **(D)** HFF cells were transfected with the Con-RNAi, UL84-RNAi-#2, and then infected with HCMV (MOI = 1), followed by Western blotting analysis of pSTAT1, STAT1, IRF3, pIRF3, cytoplasmic FHL2 (cy), and nuclear FHL2 (nu). **(E)** Co-IP analysis of UL84 interaction with FHL2, IRF3, c-Jun, or p300. Expression plasmids of UL84, FHL2, TBP, IRF3, p300 and c-Jun were transiently transfected into HEK293T cells for 48 h, and then the total cell lysates were prepared

and immunoprecipitated with anti-UL84, anti-FHL2, anti-IRF3, anti-c-Jun or anti-p300 antibodies, followed by immunoblot analysis. **(F)** Co-IP analysis of UL84 (Δ400-460) interaction with FHL2, IRF3, c-Jun, or p300. Expression plasmids of UL84(Δ400-460), FHL2, TBP, IRF3, p300 and c-Jun were transiently transfected into HEK293T cells for 48 h, and then the total cell lysates were prepared and immunoprecipitated with anti-Myc, anti-FHL2, anti-IRF3, anti-c-Jun or anti-p300 antibodies, followed by immunoblot analysis. **(G)** HEK293T cells were transfected with pCDNA3.1-FHL2, pCDNA3.1-IRF3, pCDNA3.1-p300, and pCDNA3.1-c-Jun plasmids, along with increasing amounts of the pCDNA3.1-UL84 expression vector for 48 h. Co-IP analysis of UL84 interaction with FHL2, IRF3, c-Jun, or p300. **(H)** Effect of different mutations on the interaction of FHL2 with UL84. HEK293T cells were transiently transfected with Flag-tagged FHL2 (four single point mutants) and pcDNA3.1-UL84 for 48 h before Co-IP analysis. **(I)** Co-IP analysis of UL84 interactions with TBP. HEK293T cells were transiently transfected with the indicated plasmids for 48 h, using as the indicated anti-UL84, anti-Flag or anti-TBP antibody before Co-IP analysis was performed. **(J)** Molecular docking simulation of the complex of UL84-FHL2-TBP. The structure of the IRF3, FHL2 and TBP was downloaded from the Protein Data Bank, green refers to FHL2, grey refers to TBP, and blue refers to UL84. The displayed images were representative ones from three independent experiments. For all figures, statistical analyses were performed using two-tailed t-test. Differences were considered statistically significant when * denoted $p < 0.05$, ** denoted $p < 0.01$, *** denoted $p < 0.001$, and **** denoted $p < 0.0001$.

## UL84 interacts with FHL2 to help HCMV evade the innate immune response and promotes viral *ori*Lyt-dependent DNA replication

Next, we further confirmed the importance of FHL2 in activating the transcription of IFN-β and its corresponding biological effect on virus growth, as well as the role of UL84 in confrontation. We first examined the mRNA levels of *IFNB1* by qPCR analysis during HCMV infection of HFFs ([Fig 9A]). The results indicated that overexpression of FHL2 or interference of UL84 by specific RNAi significantly increased the expression of *IFNB1* at mRNA level after HCMV infection. In contrast, interference of FHL2 by RNAi or overexpression of UL84 dramatically decreased the level of interferon in HCMV-infected cells. Meanwhile, once FHL2 was down-regulated or mutated to FHL$^{T112G}$, overexpression of UL84 did not further inhibit interferon levels, indicating that the regulation of UL84 on *IFNB1* expression acted through its interaction with FHL2. Accordingly, we can see the opposite trend that overexpression of FHL2 greatly suppressed the virus titer, while cells with FHL2 knock-down or overexpressed with FHL$^{T112G}$ infected by HCMV showed a significant increase in viral titer compared to Con-RNAi transfected group ([Fig 9B]). Also, overexpression of UL84 could effectively counter the inhibitory effect of FHL2, and the enhancement was not further enhanced by the down-regulation of FHL2.

It was reported that UL84 is required *ori*Lyt-dependent DNA replication, but very little is known about the mechanism of UL84 in this process [32]. To investigate whether the interaction between UL84 and FHL2 directly affects viral DNA replication, a transient transfection-replication reporter assay previously described by our group and others was adopted in this study [57,58]. Results indicated that ectopically overexpressing FHL2 dramatically inhibited the activation of the *ori*Lyt-replication reporter, while down-regulating FHL2 by RNAi considerably activated the replication of *ori*Lyt compared to minimal *ori*Lyt-replication system (all rep + IE2 + UL84 + *ori*Lyt group) ([Fig 9C]). The overexpression of FHL2$^{T112G}$, which lost the interaction with UL84, however, did not inhibit *ori*Lyt DNA synthesis.

These results indicate that UL84 is essential for evasion of innate antiviral response mediated by host endogenous FHL2, and the interaction is antagonistic in the process of HCMV DNA replication.

## Discussion

In this study, we demonstrated that phosphorylation of FHL2 serves as part of the interferon-activated cascade, and FHL2 recruits TBP to promote the initiation of IFN-β transcription. Meanwhile, HCMV UL84 antagonists the IFN-I responses by targeting FHL2 and disrupts the FHL2-mediated assembling of enhanceosome complex.

The role of FHL2 as an adapter protein in facilitating protein-protein interactions has been established. Through the examination of the LIM domain of the FHL2 protein, we discovered its critical function in activating IFN-β transcription and enhancing antiviral innate immunity. In the context of viral infection and the subsequent activation of innate immunity, it was observed that UL84, the HCMV replication initiation protein, exploits FHL2 to counteract its mediated innate antiviral response. During *ori*Lyt-dependent DNA replication, the interaction between these two factors is antagonistic.

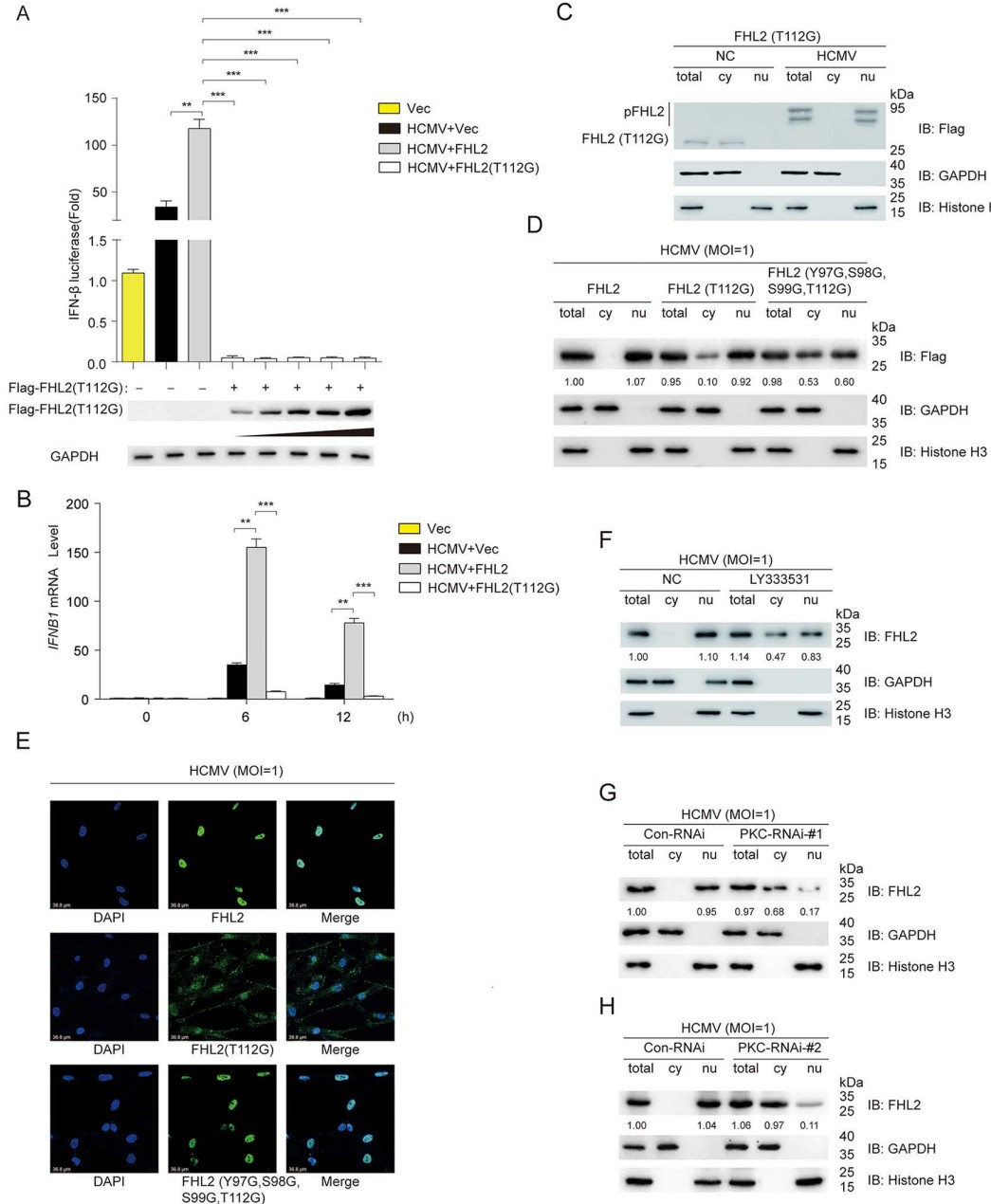

**Fig 8. T112 plays a role in interferon-activated transcription. (A)** FHL2 (T112G) inhibits the activation of the IFN-β promoter in a time-dependent manner. pGL3-luci-IFN-beta reporter plasmids and pcDNA3.1-Flag-FHL2 (T112G) expression plasmids were transiently transfected into HFFs for 24 h before being infected with HCMV at a multiplicity of infection (MOI = 1). Subsequent luciferase assays were conducted at various time points following infection. **(B)** HFF cells were transfected with the plasmid of pcDNA3.1-FHL2, pcDNA3.1-FHL2(T112G) or the empty vector for 24 h, and then infected with HCMV (MOI = 1) for the indicated times before RT-qPCR analysis of the indicated antiviral interferon-stimulated genes. **(C)** Phosphorylation of Flag-tagged FHL2(T112G) under HCMV infection by immunoblot and Phos-tag analysis. HCMV (MOI = 1) infected HFFs for 6 h, prepared HFF cells and isolate cytoplasmic and nuclear proteins using the Nuclear and Cytoplasmic Protein Extraction Kit before Phos-tag SDS-PAGE and immunoblot analysis. **(D)** Effect of different mutations of FHL2 on its nuclear translocation. HFFs cells were transiently transfected with Flag-tagged FHL2 (full-length or a single point mutant (T112G) or a four-point mutant (Y97G, S98G, S99G, T112G)) for 48 h before Co-IP analysis. **(E)** The colocalization of FHL2, FHL2 (T112G), and FHL2 (Y97G, S98G, S99G, T112G) during HCMV infection was investigated. HFFs were transiently transfected with pcDNA3.1-FHL2, pcDNA3.1-FHL2 (T112G), and FHL2 (Y97G, S98G, S99G, T112G) for 24 h before being infected with HCMV (MOI = 1). After infection, the cells were fixed and permeabilized, followed by incubation with primary antibodies against FHL2 or Flag, and subsequently with Dylight 647-conjugated fluorescent

secondary antibodies. The subcellular localization of FHL2 (indicated in red) and DAPI (indicated in blue) was examined using confocal microscopy. **(F)** Effect of PKC inhibitor LY333531 on endogenous FHL2 nuclear translocation under HCMV infection. HFFs were treated with LY333531 (200 nM) for 24 h, followed by infection with HCMV (MOI = 1) for 6 h. Subsequently, immunoblot analysis was performed to assess FHL2 nuclear translocation. **(G) (H)** HFFs were treated with PKC-RNIAi-#1 or PKC-RNIAi-#2 for 24 h, followed by infection with HCMV (MOI = 1) for 6 h. Subsequently, immunoblot analysis was performed to assess FHL2 nuclear translocation. The displayed images were representative ones from three independent experiments. For all figures, statistical analyses were performed using two-tailed t-test. Differences were considered statistically significant when * denoted $p < 0.05$, ** denoted $p < 0.01$, *** denoted $p < 0.001$, and **** denoted $p < 0.0001$.

FHL2 is involved in various biological processes, including cell proliferation, apoptosis, cell differentiation, and tissue repair. Previous studies have indicated that FHL2 plays a role in the transcriptional regulation of genes associated with these processes. For instance, FHL2 has been shown to inhibit estrogen receptor (ER) activity by enhancing the interaction between estrogen receptor α (ERα) and the maternal anti-apoptotic protein Smad4. However, the specific mechanisms through which FHL2, with its multifaceted functions, participates in gene transcription regulation remain unclear. Our study demonstrated that the phosphorylation of FHL2 can act as part of a signaling cascade triggered by DNA viruses or RNA viruses. FHL2 predominantly resides in the cytoplasm; however, upon viral infection, immunofluorescence confocal experiments revealed that FHL2 translocated entirely to the nucleus. Given that FHL2 is associated with the transcriptional activation of c-Jun, AP-1, EGF, EGFR, TCF/LEF, and others, yet does not directly bind to nucleic acids, we speculate that FHL2 may function as an activator in the transcriptional activation of IFN-β. To verify this hypothesis, chromosomal co-immunoprecipitation experiments confirmed that FHL2 is indeed involved in the activation of the IFN-β promoter, consistent with previous findings. It is well established that a primary function of FHL2 is to enhance protein-protein interactions. Since FHL2 was previously reported to interact with c-Jun, we hypothesized that FHL2 facilitates the formation of enhanceosome complexes, thereby upregulating IFN-β transcription. The results of co-immunoprecipitation experiment further demonstrated that FHL2 exhibits a strong interaction with enhancer elements IRF3, p300, and c-Ju. Notably, we identified that the key domains recognized by FHL2 for its interactions with IRF3, p300, and c-Jun are all LIM2, with critical amino acid positions at 97Y, 98S, 99S, and 112T. Previous studies on FHL2 and transcription factors have primarily concentrated on the LIM2 or LIM3 domains; however, the mechanism by which these domains promote protein-protein interactions remains inadequately elucidated. Investigations into the upstream elements of the IFN-β promoter have revealed a spatial gap of more than ten base pairs between the enhancer complex and TFIID. Therefore, we hypothesize that the FHL2 protein may also act as an adapter protein, facilitating the recruitment of TFIID to initiate transcription. Our study demonstrates that FHL2 binds to enhancer complexes via LIM2 recognition in its N-terminal segment, while LIM3, LIM4, or the combination of LIM3-LIM4 recognition facilitates the recruitment of TBP to the TATA box, thereby activating transcription initiation. The LIM domain is characterized by a series of cysteine residues, and mutations at positions 97Y, 98S, 99S, and 112T do not appear to disrupt this interaction. Consequently, we speculate that TBP recognition and binding to the LIM domain may be mediated by a cysteine-rich sequence "CX2CX2C". To validate this hypothesis, further experimental investigations are warranted.

Cellular proteins that contain LIM domains are widely found in eukaryotes. The human genome encodes 58 genes containing LIM domains [59]. LIM-domain-only protein family members have no NLS (nuclear localization signal) or NES (nuclear export signal) domains, but usually have the function of PPI. NLS is usually regulated by phosphorylation, and the amino acid residues adjacent to the NLS peptide are modified by phosphorylation [60,61]. Phosphorylation induces a small change in protein structure to another activated status, promoting PPI [62]. Therefore, the LIM-domain-only protein family members have the function that shuttles from the nucleus to the cytoplasm, which may be a key factor in the induction of post-translational phosphorylated modification. In addition, phosphorylation of amino acid residues at single or multiple sites can also confer the proteins with diverse functions. For example, the amino acids Y97, Y176, Y217, Y236, and S238 of FHL2 that undergo phosphorylation modifications are involved in the cell cycle regulation [50,55,56]. To sum

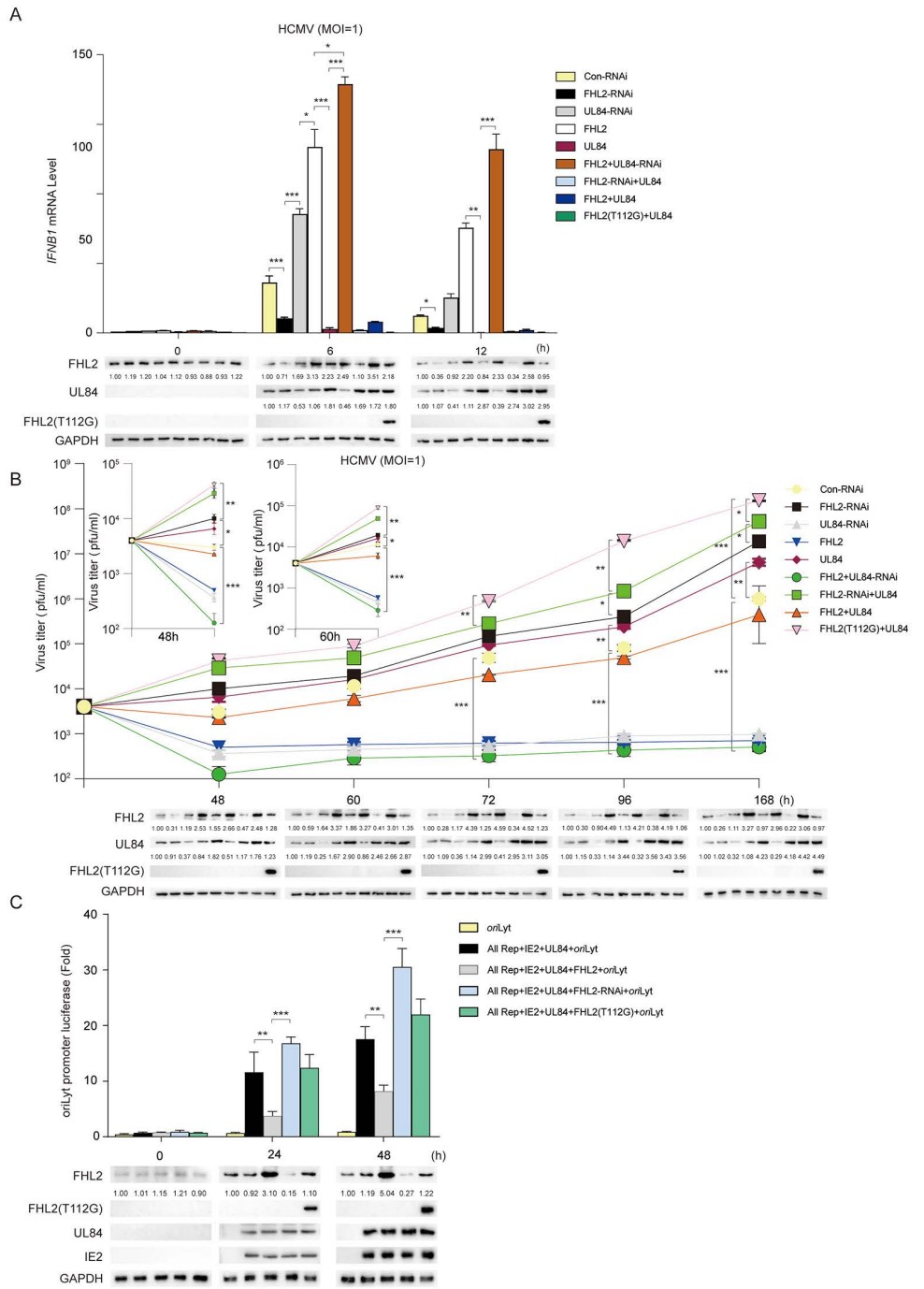

**Fig 9. UL84 interacts with FHL2 to help HCMV evade the innate immune response. (A)** Con-RNAi, FHL2-RNAi, UL84-RNAi, FHL2, UL84, FHL2 with UL84-RNAi, FHL2-RNAi with UL84, FHL2 with UL84, and FHL2(T112G) with UL84 were transiently transfected in HFFs for 24 h, respectively. Then infected with HCMV (MOI = 1). Samples were harvested at different time points. One portion of the samples was analyzed for IFN-B1, normalized against GAPDH mRNA levels before RT-qPCR analysis. Another portion was subjected to Western blotting. **(B)** Effects of FHL2 on the replication of HCMV. Con-RNAi, FHL2-RNAi, UL84-RNAi, FHL2, UL84, FHL2 with UL84-RNAi, FHL2-RNAi with UL84, FHL2 with UL84, and FHL2(T112G) with UL84 were transiently transfected in HFFs. After transfection, HFFs were infected with HCMV-WT. Supernatants were harvested at 48, 60, 72, 96, and 148 hpi for measurement of viral titers. Another portion was subjected to Western blotting. **(C)** HFFs were transiently transfected with pGL3-Luc-oriLyt, pCDN3.1-IE2, pCDN3.1-UL84, pCDN3.1-Flag-FHL2 (T112G), pCDN3.1-FHL2 and FHL2-RNAi for 24 h, HCMV (MOI = 1) reinfection of cells for 6 h or 12 h

before luciferase assay and Western blotting. The displayed images were representative ones from three independent experiments. For all figures, statistical analyses were performed using two-tailed t-test. Differences were considered statistically significant when * denoted $p < 0.05$, ** denoted $p < 0.01$, *** denoted $p < 0.001$, and **** denoted $p < 0.0001$.

up, post-translational modifications can endow LIM-domain-only proteins with diverse functions such as transcriptional activation, signal transduction, and cell cycle regulation. Phosphatomics and proteomics linkage studies will further reveal the causes and mechanisms of functional diversity of LIM-domain-only proteins.

In addition, the IFN-β enhancer has a remarkable property that controls the transcription very precisely. TBP and TAF$_{II}$250 (TFIID) were recruited to the promoter 6 h post infection (hpi) and the nucleosome II was remodeled at 9–19 hpi [40]. We found that when HFFs were infected with HCMV, FHL2 had completely entered the nucleus at 6 hpi. The timing of FHL2 involvement in IFN-β transcription coincides with the recruitment of TBP to the TATA-box and nucleosome II remodeling. Therefore, we demonstrated that FHL2 provides a PPI interface as a bridge with histone acetyltransferase attached at the N-terminal and TBP recruited at the C-terminal, thus, to promote the initiation of IFN-β transcription. Interestingly, mutation of any amino acids in Y97, S98, S99, or T112 in FHL2 abolished the ability to interact with IRF3, c-Jun, and p300. T112 is also a key site for UL84 to recognize FHL2, and FHL2 $^{T112G}$ does not activate IFN-β transcription. We hypothesized that the amino acid cluster within FHL2 functions as a transcriptional synergist and has a potential to be a cellular target for drug design.

It was reported that UL84 is required *ori*Lyt-dependent DNA replication *in vitro* of strains AD169 or Towne [63,64], but very little is known about the mechanism of UL84 in this process [32]. To investigate whether the interaction between UL84 and FHL2 affects viral DNA replication, a transient transfection-replication reporter assay was adopted in this study. Results indicated that ectopically overexpressing FHL2 dramatically inhibited the activation of the *ori*Lyt-replication reporter, while down-regulating FHL2 by RNAi considerably activated the replication of *ori*Lyt compared to minimal *ori*Lyt-replication system (all rep + IE2 + UL84 + *ori*Lyt group). The overexpression of FHL2$^{T112G}$, which lost the interaction with UL84, however, did not inhibit *ori*Lyt-replication reporter synthesis.

Previously, it has been shown that the N-terminal functional domain of UL84 can interact with IE2 and UL44 [30,31]. However, little is known about the C-terminal of UL84. Here we found that UL84 is a multi-functional protein that contributes to viral replication and virus-mediated immune escape. We determined the mechanism by which UL84 enters the nucleus and help HCMV evade innate immune response. UL84 is phosphorylated and translocated into the nucleus. The C-terminus of UL84 competitively and reversibly binds to FHL2 through the T112 site, forming a UL84-FHL2 complex and disrupt the formation of the IRF3-c-Jun-p300 complex, thus prevent FHL2-mediated recruitment of TBP to TATA-box. Ectopic expression of UL84 had dramatic inhibitory effects on the expression of IFN-β stimulated genes induced by not only HCMV but also other DNA viruses such as HSV, as well as RNA viruses such as IAV, EV71 and ZIKV.

On the other hand, it was interesting that UL84 is an origin-binding protein (OBP) of viral replication involved in innate immune escape. It was shown that IE2 of HCMV interacts with TBP, SP1, c-Jun, and p300/CREB, but the function is yet unknown, which is possible through an indirect interaction [65]. However, UL84 can modulate IE2-mediated repression of HCMV *ori*Lyt$_{PM}$ [31]. The results indicated an indirect interaction between UL84 and TBP, and the function of UL84 during HCMV replication may be to recruit the transcription factor TFIID to the TATA-box. Nonstructural proteins (e.g., HCMV UL84) act as bridges to connect PIC-transactivators with GFTs to switch gene transcription from a repressed to an activated state. Interestingly, deletion of UL84, according to previously reported in the TB40 strain, does not seem essential for viral replication [63], although it can still affect the viral replication efficiency to some extent. This observation strongly supports the viewpoint of this study that UL84 has functions beyond its traditional role as a protein considered essential for replication. In the subsequent study, we will conduct a more in-depth exploration of the mechanism by which UL84, beyond its function in inhibiting the transcription of IFN-β via interaction with FHL2, facilitates oriLyt-dependent DNA

synthesis. This study lays a theoretical and practical foundation for developing specific antiviral drug candidates that target the dual biological functions of UL84.

## Materials and methods

### Cells and viruses

Human foreskin fibroblasts (HFF cells, passages 9–14), HEK293T and human lung epithelial lung cell line A549 were obtained from the American Type Culture Collection (ATCC) (Manassas, VA, USA). HCMV (Towne-BAC) were kindly provided by Prof. Fenyong Liu (University of California, Berkeley, CA, USA). A/WSN/34 H1N1 were provided by Dr. Wenjun Song (Guangzhou Medical University). HSV, EV71 (Xiangyang-Hubei-09) and ZIKV (z16006) strains were kindly provided by Dr. Zhen Luo (Jinan University, Institute of Medical Microbiology).

### Reagents and antibodies

Anti-Flag mouse monoclonal antibody (Cat. No. F1804; diluted 1:10,000) antibody and Monoclonal FHL2 mouse monoclonal antibody (Cat. No. WH0002274M1; diluted 1:10,000) was purchased from Sigma (Saint Louis, USA). The antibodies against human IRF3 rabbit monoclonal antibody (Cat. No. 11904; diluted 1:1,000), human pIRF3 rabbit monoclonal antibody (Cat. No. E6F7Q; diluted 1:1,000), c-Jun mouse monoclonal antibody (Cat. No. 2315; diluted 1:1,000), STAT1 rabbit monoclonal antibody (Cat. No. 9172S; diluted 1:1,000) and pSTAT1 rabbit monoclonal antibody (Cat. No. 9167S; diluted 1:1,000) were obtained from CST (Boston, USA). CMV UL84 mouse monoclonal antibody (Cat. No. sc-56977; diluted 1:1,000) was purchased from Santa Cruz Biotechnology (State of Texas, USA). Anti-c-Myc mouse monoclonal antibody (Cat. No. ab32072; diluted 1:1,000), anti-6X His tag mouse monoclonal antibody (Cat. No. ab5000; diluted 1:1,000), anti-GST rabbit monoclonal antibody (Cat. No. ab111947; diluted 1:1,000), Rabbit Recombinant Monoclonal FHL2 antibody (Cat. No. ab202584; diluted 1:1,000), anti-GAPDH rabbit monoclonal antibody (Cat. No. ab181602; diluted 1:10,000), anti-KMT6/EZH2 rabbit monoclonal antibody (Cat. No. ab307646; diluted 1:1,000), anti-Histone H3 rabbit polyclonal antibody (Cat. No. ab1791; diluted 1:5,000), anti-KAT3B/p300 rabbit polyclonal antibody (Cat. No. ab10485; diluted 1:5,000), anti-TATA binding protein TBP rabbit monoclonal antibody (Cat. No. ab220788; diluted 1:1,000), anti-Cytomegalovirus IE1 and IE2 mouse monoclonal antibody (Cat. No. ab53495; diluted 1:1,000), anti-Influenza A Virus Nucleoprotein mouse monoclonal antibody (Cat. No. ab128193; diluted 1:300), Goat Anti-Rabbit IgG H&L (Cat. No. ab150079; diluted 1:200), Goat Anti-Mouse IgG H&L (Alexa Fluor 568) (Cat. No. ab175473; diluted 1:200), and anti-HSV 1 ICP4 Immediate Early Protein (Cat. No. ab6514; diluted 1:800) were purchased from Abcam (Cambridge, UK). Zika virus NS1 mouse monoclonal antibody (Cat. No. EA88; diluted 1:50) was purchased from Invitrogen (California, USA). Enterovirus 71 VP1 rabbit polyclonal antibody (Cat. No. GTX132338; diluted 1:1,000) was obtained from GeneTex. DAPI (Shanghai, China; Cat. No. C1002) was purchased from Beyotime Biotechnology. LY333531 hydrochloride (Cat. No. HY-10195) was purchased from MedChemExpress (New Jersey, USA).

### Plasmids design and construction

Genes encoding UL84, IE2, UL44 were amplified from the HCMV Towne and subcloned into the expression vector, respectively. Plasmids encoding FHL2, EZH2, p300, IRF3, c-Jun and TBP were generated by PCR amplification using genomic DNA isolated from normal human HFF cells as the template. All plasmids including those previously constructed and preserved in our laboratory were verified by sequencing to confirm accuracy. The primer pairs used for plasmid construction are listed in S2 Table.

Plasmid map sequence information of pCDNA3.1(GenBank: MN996867.1), a mammalian expression vector with a CMV promoter. Plasmid map sequence information of pCMV-N-myc (Addgene, Cat. No. 631604), a mammalian expression vector with a CMV promoter. Plasmid map sequence information of pRK11-Flag (Cat. No. P33800), a mammalian expression vector with a CMV promoter was purchased from MIAOLING PLASMID (Wuhan, China). Plasmid map sequence

information of pGEX-6P-1-N-GST (Catalog Number: VT1258), a mammalian expression vector with a Tac promoter was purchased from YouBio (Changsha, China). Plasmid map sequence information of pET28A-N-His (Cat. No. P0023), a mammalian expression vector with a T7 promoter was purchased from MIAOLING PLASMID (Wuhan, China). Plasmids encoding FHL2 with site-specific mutations were constructed via the overlap extension PCR method listed in S2 Table.

The pGL3-luci-IFN-beta and pGL3-luci-NF-κB plasmid provided by Prof. Yonggang Pei (Southern University of Science and Technology, Shenzhen, China). pGL3-luci-IFN-β contains two interferon-stimulated response element (IRSE) repeats, and each core element harbors conserved sequences recognized by c-Jun (consensus sequence: 5'-TGACATAG-3'), IRF3 (consensus sequence: 5'-AANNGAAA-3'), and p65 (consensus sequence: 5'-GGGRNNYCC-3'). Sequencing results of pGL3-luci-NF-κB revealed two 10 bp κB-responsive promoter consensus sequences (GGGACTTTCC) located 51 bp and 76 bp upstream of the TATA box.

UL84 synonymous mutations was performed using the mutation primers and the Fast Site-Directed Mutagenesis Kit (Transgen Biotech, Beijing, China; Cat. No. FM111-01) according to the manufacturer's protocol. The mutation primers are as follows:

UL84-mut-F: TCGGGTGCACCGGGGCACCTACCACCTCATTCAGTTGCAC,

UL84-mut-R: GTGCAACTGAATGAGGTGGTAGGTGCCCCGGTGCACCCGA.

## HFF cell transfection

HFF cells were plated in 10 cm² dishes and achieved 90% confluency one day before electroporation [66,67]. Pre-electroporation, cells were detached with trypsin, resuspended in electroporation buffer (Bio-Rad, USA; catalog number 1652676) to a concentration of 2–5 × 10⁶ cells/mL. Two hundred microliters of cell suspension were combined with 1 μg plasmid in an electroporation cuvette, followed by electroporation using a Gene Pulser Xcell system set at square wave, 160 V, 15 ms [66,67]. Immediately after electroporation, the cell suspension was transferred to 6 cm culture dishes for incubation. The medium was aspirated and refreshed after 4 hours. Transient gene expression was assayed 24–48 hours post-electroporation.

## Transfection-Replication Assays

Construction of the replication reporter plasmid pGL3–14 and replication-related plasmids was described in our previous work [57]. HFF cells were trypsinized, resuspended in electroporation buffer, mixed with 10 μg total DNA (HCMV core replication proteins/cofactors-encoding plasmids) and 1 μg pGL3–14 reporter plasmids, and electroporated [57,66,67]. Immediately after electroporation, the cell suspension was transferred to culture dishes for incubation. The medium was aspirated and refreshed after 4 hours. After 24 hours, cells were harvested and *ori*Lyt promoter-driven reporter activation levels were assayed. (Beyotime, Shanghai, China; catalog number RG029M).

## siRNA Transfection

Human foreskin fibroblast (HFF) cells were cultured in culture plates, and siRNA transfection was performed using StarvioPM siRNA/miRNA Transfection Reagent (Cat#: 11013, Baidai Biotechnology, Changzhou, China) in accordance with the manufacturer's instructions. Transient gene expression was detected 24–48 hours post-transfection. siRNA targeting protein kinase C-δ (si-PKC-δ) and si-UL84 were purchased from Sangon Biotech (Shanghai, P.R. China). The sequences of siRNA used in this study are as follows:

UL84-siRNA-1-F: 5'-CGCGGAACUUACCAUCUAAUC-3',

UL84-siRNA-1-R: 5'-UUAGAUGGUAAGUUCCGCGGU-3';

UL84-siRNA-2-F: 5'-CGCGGAACUUACCAUCUAAUC-3',

UL84-siRNA-2-R: 5'-UUAGAUGGUAAGUUCCGCGGU-3';

PKC-siRNA-1-F: 5'-CAAGAAGUGUAUUGAUAAAGU-3',

PKC-siRNA-1-R: 5'-UUUAUCAAUACACUUCUUGUG-3';

PKC-siRNA-2-F: 5'-GAUUUAAAGUCUACAAUUACA-3',

PKC-siRNA-2-R: 5'-UAAUUGUAGACUUUAAAUCUG-3'.

## Establishment of stable cell lines

Search and use the online website http://crispr-era.stanford.edu/index.jsp. Design the sgRNA according to the operation instructions, then construct it into the LentiCRISPR V2 vector. The primer sequences are as follows:

FHL2-1: ACUGAGCGCUUUGACUGCCACCAUU [68],

FHL2-2: CGAAUCUCUCUUUGGCAAG [69],

FHL2-3: UCUCUCUUUGGCAAGAAGU [69];

Construction of knockout plasmids was performed by constructing and transforming the knockout plasmids into Stabl3 competent cells. Following plasmid extraction, lentivirus packaging and concentration were conducted. A mixture of 8 µg/mL polybrene and 50 µL concentrated lentivirus solution was incubated at room temperature for 30 minutes. After incubation, the mixture was added dropwise to a 6-well plate, which was then placed in an incubator for culture. After 48 hours, the medium supernatant was discarded, and puromycin was added to achieve a final concentration of 5 µg/mL. The plate was subsequently returned to the incubator for continued culture. Every two days, the medium was replaced with fresh complete medium containing 5 µg/mL puromycin for further selection, and the process lasted for two weeks. Selected cells were then cultured for expansion and identification before experiments.

## Yeast two-hybrid screening and assays

*Saccharomyces cerevisiae* strain AH109 control vectors pGADT7, pGADT7-T and pGBKT7-p53, and a human fetal brain cDNA library were purchased from was purchased from Clontech (Mountain View, CA, USA). Prior to library screening, pGBKT7-UL84 (GAL4 DNA-binding domain [GAL4-BD]-UL84 fusion plasmid) was transformed into AH109 to exclude UL84-mediated self-activation of yeast reporter genes. Subsequently, AH109/pGBKT7-UL84 was co-transformed with cDNA library plasmids. Transformants were plated on synthetic dropout medium lacking Trp, Leu, Ade, and His (SD/-Trp/-Leu/-Ade/-His) and incubated at 30°C for 3–5 days. Visible colonies were validated by β-galactosidase activity assays (5-bromo-4-chloro-3-indolyl-β-D-galactopyranoside [X-Gal] filter lift method). Plasmids from positive clones were extracted using the E.Z.N.A. Yeast Plasmid Kit (Omega Bio-Tek), rescued in Escherichia coli DH5α competent cells, and purified via standard miniprep. For the negative control, AH109 was co-transformed with empty pGBKT7 and the cDNA library, following the same procedures. Inserts of positive plasmids were sequenced with the GAL4AD-specific primer (5'-TAATACGACTCACTATAGGGC-3'), and homology was analyzed via BLAST on the NCBI database to identify target genes [70].

## Luciferase reporter assays

A dual-luciferase reporter assay system was used to assess that whether FHL2, FHL2$^{T112G}$, UL84 activate or repress the IFN-β promoter. The plasmids were co-transfected into HEK293T cells alongside pGL3-luci-IFN-beta and pRL-TK. After

24 hours, the cells were infected with HCMV (MOI = 1). Subsequently, at 6 hours post-infection (hpi), cells were harvested and promoter-driven reporter activation levels were assayed (Beyotime, Shanghai, China; catalog number RG029M).

## SDS-PAGE and immunoblot analysis

Cells were lysed by adding lysis solution (20 mM Tris-HCl pH7.4, 150 mM NaCl, 4% detergent, 2mM EDTA, 10% glycerol) for 30min at 4°C rotating, after then, were centrifuged and taken the supernatant removal. The supernatant was mixed with 6 × SDS-PAGE loading buffer (1.5M Tris-HCl pH7.4, 10% glycerol, 2% SDS, 15% β-mercaptoethanol, 0.0005% bromophenol blue) and boiled for 10min. The protein samples were separated by SDS-PAGE and transferred to PVDF (Millipore, New Jersey, USA; catalog number IPFL00010) membrane. After immunoblotting, the PVDF membrane were incubated in the blocking buffer (3% nonfat dry milk and 0.05% Tween-20 in TBS) for 1h. After that, the membrane was transferred to the primary antibody dilution for 1h at room temperature, followed by 3 times washes with TBST. After the membrane were washed by TBST, the membrane is transferred to the secondary antibody dilution and incubated for 1h at room temperature, followed by 3 times washes with TBST. After the membrane were washed by TBST, the results are displayed using chemiluminescence.

## GST/His pull-down assay

Construction of fusion protein expression plasmids: pGEX-6P-1-FHL2, pGEX-6P-1-EZH2, pET28a-UL84, pET28a-UL44. And pGEX-6P-1-EZH2 and pET28a-UL44 vector served as negative controls. For GST/His-tag pull-down, GST/His--tagged recombinant proteins were mixed with glutathione or anti-His at 4°C and incubated with Protein A+G agarose (Beyotime, Shanghai, China; catalog number P2019) for 4h. After washing the agarose three times with PBS, mixed 6 × SDS-PAGE loading buffer and boiled for 10min. Then SDS-PAGE with immunoblot analysis assay pull-down results.

## Co-immunoprecipitation assay

After transfection of plasmid being into HEK293T cells for 48h, cells were collected with 200 µl lysis solution and 2 µl protease inhibitor, then incubated for 30min at 4°C, and then the supernatant was removed by centrifugation. The supernatant was added with 30 µl Protein A+G agarose, 2 µl protease inhibitor, 0.8 µl primary antibody and incubated for 2h at 4°C. After incubation, the beads were washed 3 times with pre-cooled PBS and the supernatant was discarded. The beads were added with 25 µl of lysis solution and 5 µl loading buffer (6 × SDS-PAGE loading buffer) and boiled for 10min. Immunoblot analysis of interaction results.

## Indirect immunofluorescence assay and confocal microscopy

First, the confocal dish is rinsed with phosphate-buffered saline (PBS), following which 4% paraformaldehyde is added and the dish is incubated at room temperature for 20 minutes to fix the samples. Upon completion of fixation, the confocal dish is washed three times with PBS to remove residual fixative, then PBS-T (PBS supplemented with 0.2% Triton X-100) is added and incubated for 10 minutes to permeabilize the cell membranes. After permeabilization, the dish is again washed three times with PBS to eliminate excess PBS-T, and subsequently PBS-B (PBS containing 4% bovine serum albumin, BSA) is added for incubation at 37°C for 10 minutes to block non-specific binding sites. Following the blocking step, primary antibody is added to the dish and incubated at room temperature for 2 hours to allow specific binding to the target antigen. Next, the dish is washed three times with PBS to remove unbound primary antibody, after which secondary antibody is added and incubated at room temperature for 1 hour to bind to the primary antibody. Once the secondary antibody incubation is finished, the dish is washed three times with PBS to discard unbound secondary antibody, and DAPI staining solution is then added for nuclear staining. Finally, the prepared confocal dishes are scanned and observed using a confocal laser scanning microscope to acquire imaging data.

## Phos-tag analysis

Detect FHL2 phosphorylation by in vitro Phos-tag gel, and there are two main concerns in sample preparation protocol: (1) Cytoplasmic samples: HFF cells were lysed in lysis buffer (10mM Tris-HCl [pH7.4], 150 mM NaCl, 1% NP-40, 0.25% sodium deoxycholate, 2mM EGTA) containing protease inhibitor and phosphatase inhibitor for 30 min at 4°C. The supernatant was then removed by centrifugation at 12000 g for 10 min at 4°C, and the supernatant was added to protein loading buffer (1.5M Tris-HCl [pH 7.4], 10% glycerol, 2% SDS, 15% β-mercaptoethanol, 0.0005% bromophenol blue) containing 2 µl 10 µM $MnCl_2$ was boiled for 10 min. (2) Nucleus samples: centrifuged precipitates were added to lysis buffer (10mM Tris-HCl [pH7.4], 150 mM NaCl, 1% NP-40, 0.25% sodium deoxycholate, 2mM EGTA) containing protease inhibitor and phosphatase inhibitor and then sonicated for 2 min. Then the supernatant was removed by centrifugation at 12,000 g at 4°C for 10 min. The supernatant was removed for 10 min, and the supernatant was added to protein loading buffer (1.5M Tris-HCl, 10% glycerol, 2% SDS, 15% β-mercaptoethanol, 0.0005% bromophenol blue) containing 2 µl of 10 µM $MnCl_2$ and boiled for 10 min.

Cytoplasmic or nucleus samples were separated by 8% SDS–PAGE gel containing 50 µM Phos-tag acrylamide (Apex-Bio, Houston, USA; catalog number F4002) and 100 µM MnCl2. The Phos-tag gel was then washed with transfer buffer containing 1 mM EDTA to remove the $Mn^{2+}$ in three washes of 20 minutes each. And then Phos-tag gel transferred to PVDF membrane for immunoblotting.

## Molecular docking simulations

The FHL2, c-Jun, p300 and UL84 structure constructed by AlphaFold was used as a template. The FHL2, UL84 and p300 structure constructed by AlphaFold were used as a template. For IRF3, the 2O61 crystal at 2.80 Å resolution was selected. For TBP, the 4ROE crystal at 2.20 Å resolution was selected. The protein structure was prepared with "Protein Preparation Wizard" (Schrödinger, LLC, New York, NY2021). application using default settings, adding hydrogens, assigning disulfide bonds, removing surrounding waters, adjusting charges, capping termini, and adding missing side chains using Prime. The protein–protein docking experiments were performed using the Protein-Protein Docking component. All other parameters were kept default. The best pose was output on the basis of prime energy and the protein-protein interactions.

Molecular dynamics simulations of the near-native docked FHL2 structural and four mutants computationally constructed were executed to evaluate the stability and dynamic behavior of proteins. For the stability analysis following molecular dynamics simulations, six systematic analyses were conducted: (1) Radius of Gyration (Rg) was utilized to evaluate the overall structural compactness; (2) Root Mean Square Deviation (RMSD) was calculated to primarily assess the overall drift of protein and ligand structures relative to the reference structure; (3) Root Mean Square Fluctuation (RMSF) was employed to examine the flexibility of individual protein residues; (4) Solvent Accessible Surface Area (SASA) was measured to determine the exposure level of hydrophobic regions of the protein, which is pertinent to protein folding and interactions; (5) Free Energy Landscape (FEL) analysis was performed to characterize the energy barriers and stable states associated with protein conformational transitions; and (6) Hydrogen Bond Analysis (Hbond) was conducted to quantify the number and stability of hydrogen bonds either between the protein and ligand or within the protein itself.

## Chromatin immunoprecipitation assay (ChIP)

HFF cells were infected with HCMV (MOI = 1) for 6 h, and then cross-linked with 1% formaldehyde for 15 min. Then add 125mM glycine and incubate for 5 min at room temperature. The cells were collected and centrifuged to discard the supernatant, and the lysate was added to sonicate to break the genome about 500 bp. The supernatant was collected by centrifugation and added to agarose (Beyotime, Shanghai, China; catalog number P2108). Anti-TBP/anti-FHL2 was set as the experimental group, no antibody control, rabbit IgG was the negative control, anti-RNA polymerase II was the

positive control, and incubated at 4°C overnight. The agarose was washed with low-salt wash buffer, high-salt wash buffer, and LiCl wash buffer, TE Wash buffer, respectively. Wash buffer was then added to the agarose and the supernatant was collected by centrifugation. 5 M NaCl was added to the supernatant to de-crosslink, then RNase A was added, and the RNA was removed by incubating at 37°C for 1 h. The DNA fragments were then recovered, and the enrichment of IFN-β promoter was detected by Q-PCR. The sequence of IFN-β promoter used in this study are as follows:

P1:(F-5′-TCGTTTGCTTTCCTTTGCTT-3′,

R-5′-CCCACTTTCACTTCTCCCTTT-3′) [ [71];

P2:(F-5′-AAAGGGAGAAGTGAAAGT-3′, R-5′-TACTACCTGTTGTGCCAGAGC-3′) [71,72];

P3:(F-5′-GCTCTGGCACAACAGGTAGTA-3′, R-5′-TGTAGTGGAGAAGCACAACAGG-3′) [72].

## RT-qPCR analysis

Total RNA was extracted from cell beads using a total RNA kit (Yasen, Shanghai, China; Cat. No. 19221ES50). The extracted RNA samples had an OD260/OD280 optical density ratio of 1.9 to 2.0, and cDNA synthesis was performed using a reverse transcription kit (Vazyme, Nanjing, China; Cat. No. R223-01). For RT-qPCR, the CFX96 real-time system (Bio Rad, Hercules, CA, USA) and SYBR Green Master Mix kit (Yeasen, Shanghai, China; Cat. No. 11201ES08) were used to detect the mRNA expression levels. The RT-qPCR cycling program was as follows: an initial denaturation stage of 95°C for 3 min; an amplification stage of 95°C for 20 s, 60°C for 30 s, and 72°C for 30 s for 40 cycles; and a final extension stage of 72°C for 5 min. After each run, melt curve analysis was performed to check the specificity of the amplification reaction. Using GAPDH as the data standard, calculate the relative expression of the target gene as $2^{-\Delta\Delta Ct}$. The primers used in this study were as follows:

UL84-F: CTACGCCGCTGCAATTGG;

UL84-R: GCCGCCGTTTTTTTCTCTTTG [39];

IFNB1-F: CTGGCTTCCATGAACAA;

IFNB1-F: AGAGGGCTGTGTGGAGAA [73];

ISG15-F: AGTCGACCCAGTCTCTGACTCT;

ISG15-R: CCCCAGCATTCACCTTTA [73];

IFIT1-F: GCCTAATTTACAGCAACCATGA;

IFIT1-R: TCATCAATGGATAACTCCCATGT [74];

IL6-F: CAGCTATGAACTCCTTCTCCAC;

IL6-R: GAGATGCCGTCGAGGATGTAC [75];

IRF9-F: CCACCGAAGTTCCAGGTAACAC;

IRF9-R: AGTCTGCTCCAGCAAGTATCGG [76];

TNFα-F: CAGAGGGAAGAGTTCCCCAG;

TNFα-R: CCTTGGTCTGGTAGGAGACG [77];

GAPDH-F: -TGACCTCAACTACATGGTTTACATGT;

GAPDH-R: AGGGATCTCGCTCCTGGAA [78].

## Viral titer assays

Influenza virus titration was performed using A549 cells. First, A549 cells were transfected with the GFP-UL84 plasmid via transfection reagent, respectively, one day in advance. After 24 hours, the cells were incubated with the virus for 1 hour. At the end of incubation, the cells were washed again, followed by incubation for 48–72 hours. Subsequently, the cells were fixed with 4% paraformaldehyde for 1 hour and stained with a 0.1% crystal violet solution (Sigma-Aldrich). After plaque counting, viral titers were expressed as PFU/mL [79].

HCMV titration was performed using HFF cells as previously described [57]. Briefly, to obtain the total proliferating viral samples, the infected cells as well as medium were mixed with an equal volume of 10% (w/v) skimmed milk, followed by three repeated freeze-thaw cycles and removal of cell debris by centrifugation (3000 g for 30 min at 4°C) to collect the supernatant as viral stocks. Finally, standard virus plaque assays were performed in HFF cells to measure the titers of viral stocks based on the counted numbers of viral plaques after 14-day infection. Viral titers at each time point were measured in triplicate (n = 3), with the values representing the mean of three independent experiments.

## Statistical analysis

All experiments in this study were conducted independently in triplicate, and the data are presented as mean ± standard deviation (mean ± SD). Statistical analysis of the experimental data was performed using GraphPad Prism 7 software. For comparisons between groups, an independent sample t-test (Student's t-test, $p < 0.05$) was employed for two-group comparisons, while One-way ANOVA followed by Bonferroni post hoc test was utilized for multigroup comparisons. $p < 0.05$ indicates that the difference between groups is statistically significant, * $p < 0.05$, ** $p < 0.01$, *** $p < 0.001$, **** $p < 0.0001$, ns has no significant difference.

## Supporting information

**S1 Fig. ULL84 temporal expression detection. (A)** HCMV (MOI = 1) for the indicated times before RT-qPCR analysis of the indicated *UL84* genes in HFF cells. (**B**) HCMV (MOI = 1) for the indicated times before western blot analysis of the indicated UL84, IRF3 and pIRF3 in HFF cells. All experimental assays were conducted in triplicate with independent biological replicates. For all figures, statistical analyses were performed using two-tailed t-test. Differences were considered statistically significant when * denoted $p < 0.05$, ** denoted $p < 0.01$, *** denoted $p < 0.001$, and **** denoted $p < 0.0001$. (TIF)

**S2 Fig. Confirmation that HCMV UL84 does not inhibit the NF-κB response. (A)** A549 cells were first transfected with the UL84-expressing plasmid. After 24 hours, cells were infected with IAV (MOI = 0.5). All experimental assays were conducted in triplicate with independent biological replicates. **(B) (C) (D) (E) (F)** HFF cells were transfected with pGL3-Luci-NF-κB and pRL-TK plasmids, along with increasing amounts of pcDNA3.1-UL84 expression vectors for 24 h. Then, HFFs infected with HCMV (MOI = 1), IAV (MOI = 0.5), EV71 (MOI = 1), HSV (MOI = 1), or ZIKV (MOI = 1) for an additional 24 h, after which dual-luciferase reporter assays were conducted. All experimental assays were conducted in triplicate with independent biological replicates. **(G) (H) (I) (J) (K)** HFF cells were transfected with the plasmid pcDNA3.1-UL84 or the empty vector for 24 h, and then respectively infected with HCMV (MOI = 1), IAV (MOI = 0.5), EV71 (MOI = 1), HSV (MOI = 1), or ZIKV (MOI = 1) for the indicated times before RT-qPCR analysis of the indicated TNFα genes. All experimental assays were conducted in triplicate with independent biological replicates. **(L) (M) (N) (O) (P)** HFF cells were transfected with the plasmid pcDNA3.1-UL84 or the empty vector for 24 h, then respectively infected with HCMV (MOI = 1), IAV (MOI = 0.5), EV71 (MOI = 1), HSV (MOI = 1), or ZIKV (MOI = 1) for the indicated times before RT-qPCR analysis of the indicated IL6 genes. **(Q) (R) (S) (T) (U)** HFF cells were transfected with the plasmid pcDNA3.1-UL84 or the empty vector for 24 h, then respectively infected with HCMV (MOI = 1), IAV (MOI = 0.5), EV71 (MOI = 1), HSV (MOI = 1), or ZIKV (MOI = 1) for the indicated times before RT-qPCR analysis of the indicated IRF9 genes. For all figures, statistical analyses were performed

using two-tailed t-test. Differences were considered statistically significant when * denoted p < 0.05, ** denoted p < 0.01, *** denoted p < 0.001, and **** denoted p < 0.0001.
(TIF)

**S3 Fig. Western blot analysis of HFF and HEK293T cells following transfection with plasmid. (A)** UL84-RNAi-#1 or UL84-RNAi-#2 were transiently transfected in HFFs for 24 h, respectively. HFF cells were transiently transfected via electroporation, then infected with HCMV (MOI = 1) before Western blotting. **(B)** Expression plasmids of Flag-FHL2 were transiently transfected into HEK293T cells for 48 h, and then the total cell lysates were prepared and immuno-precipitated with anti-Flag antibody, followed by immunoblot analysis. **(C)** Immunoblotting analysis was performed using anti-FHL2 antibody on lysates from the FHL2-overexpressing cell line generated in HFF cells. **(D)** Immunoblot-ting analysis was performed using anti-FHL2 antibody on lysates from the FHL2-KO cell line generated in HFF cells. **(E) (F)** PKC-RNAi-#1 or PKC-RNAi-#2 were transiently transfected in HFFs for 24 h, respectively. HFF cells were transiently transfected via electroporation, then infected with HCMV (MOI = 1) before Western blotting. **(G)** Expression plasmids of Myc-UL84 were transiently transfected into HEK293T cells for 48 h, and then the total cell lysates were prepared and immunoprecipitated with anti-Myc antibody, followed by immunoblot analysis. **(H)** Immunoblotting analysis was performed using anti-UL84 antibody on lysates from the UL84-overexpressing cell line generated in HFF cells. The displayed images were representative ones from three independent experiments. **(I)** Expression plasmids of Flag-FHL2 and Myc-UL84 were transiently transfected into HEK293T cells for 48 h, and then the total cell lysates were prepared and immunoprecipitated with anti-Flag, anti-Myc antibodies, followed by immunoblot analysis. **(J)** HFF cells were transiently transfected vector via electroporation and Lipofectamine 3000 for 48 hours. Transfection results were observed under a fluorescence microscope 48 hours after transfection. The displayed images were representative ones from three independent experiments.
(TIF)

**S4 Fig. Yeast two-hybrid screen for target proteins. (A)** The pGBK-T7-UL84 plasmid was first introduced into the AH109 strain to exclude the self-activation of reporter genes. The AH109 strain was then co-transformed with the pGBK-T7-UL84 plasmid with pGACT7-FHL2, pGACT7-KPNA3, pGACT7-KPNA4, and pGACT7-ZNF143. The AH109 strain was then co-transformed with the pGBKT7-UL84 plasmid. As a negative control, the AH109 strain was co-transformed with the pGBKT7 empty vector and the cDNA library plasmids. **(B)** Co-IP assays between UL84 and target proteins KPNA3. HEK293T cells were transiently plasmids as indicated for 48 h before Co-IP and immunoblots analysis. The displayed images were representative ones from three independent experiments.
(TIF)

**S5 Fig. FHL2 enhances virus-triggered IFN-β promoter activity and downstream interferon-stimulated gene responses. (A) (B) (C) (D)** HFF cells were transfected with pGL3-luci-IFN-beta and pRL-TK plasmids, along with increasing amounts of pcDNA3.1-FHL2 expression vectors for 24 h. Then, HFFs infected with IAV (MOI = 0.5), EV71 (MOI = 1), HSV (MOI = 1), or ZIKV (MOI = 1) for an additional 6 h, after which dual-luciferase reporter assays were conducted. All experimental assays were conducted in triplicate with independent biological replicates. **(E) (F) (G) (H)** The pGL3-luci-IFN-beta and pcDNA3.1-FHL2 plasmids were transiently transfected into HFFs for 24 h. Dual-luciferase reporter assays were then conducted. All experimental assays were conducted in triplicate with independent biological replicates. **(I) (J) (K) (L)** FHL2 was transiently transfected into HFFs for 24 h. Then IAV (MOI = 0.5), EV71 (MOI = 1), HSV (MOI = 1), or ZIKV (MOI = 1) for different times as indicated. The expression of antiviral interferon-stimulated genes (*IFIT1*, *ISG15* and *IFNB1*) was analyzed before RT-qPCR analysis, which was normalized to the mRNA level of GAPDH. All experimental assays were conducted in triplicate with independent biological replicates. For all figures, statistical analyses were performed using two-tailed t-test. Differences were

considered statistically significant when * denoted $p < 0.05$, ** denoted $p < 0.01$, *** denoted $p < 0.001$, and **** denoted $p < 0.0001$.
(TIF)

**S6 Fig. Molecular Docking Simulation. (A)** Molecular docking of the UL84-FHL2 complex: 2D interaction diagram. **(B)** Molecular docking of the FHL2-IRF3 complex. Molecular docking simulation of protein FHL2 and IRF3. The structure of the IRF3, and FHL2 were downloaded from the Protein Data Bank, green refers to FHL2, purple refers to IRF3. **(C)** Molecular docking of the FHL2-c-Jun complex. The structure of the c-Jun, and FHL2 were downloaded from the Protein Data Bank, green refers to FHL2, wine red refers to c-Jun. **(D)** Molecular docking of the FHL2-p300 complex. The structure of the p300, and FHL2 were downloaded from the Protein Data Bank, green refers to FHL2, brown refers to p300. **(E)** Molecular docking of the IRF3-FHL2-TBP complex: 2D Interaction Diagram. **(F)** Molecular docking of the UL84-FHL2-TBP complex: 2D Interaction Diagram.
(TIF)

**S7 Fig. Co-IP analysis of FHL2 interactions with TBP, IRF3, c-Jun, or p300.** Expression plasmids of Flag-FHL2, TBP, IRF3, p300 and c-Jun were transiently transfected into HEK293T cells for 48 h, and then the total cell lysates were prepared and immunoprecipitated with anti-Flag, anti-TBP, anti-IRF3, anti-c-Jun or anti-p300 antibodies, followed by immunoblot analysis. The displayed images were representative ones from three independent experiments.
(TIF)

**S1 Table. Yeast two-hybrid system screened interactions of viral proteins UL84 and Co-IP analysis.**
(DOCX)

**S2 Table. Plasmids used in this study.**
(DOCX)

## Author contributions

**Conceptualization:** Ruilin Li, Sisi Xia, Xin Li, Chuan Xia, Hongjian Li, Jun Chen.

**Data curation:** Ruilin Li, Sisi Xia, Xin Li, Jun Chen.

**Formal analysis:** Ruilin Li, Sisi Xia, Xin Li, Jun Chen.

**Funding acquisition:** Chuan Xia, Hongjian Li, Jun Chen.

**Investigation:** Ruilin Li, Sisi Xia, Xin Li, Ying Zeng, Tianqi Wang.

**Methodology:** Ruilin Li, Sisi Xia, Xin Li, Ying Zeng, Tianqi Wang, Jun Chen.

**Project administration:** Chuan Xia, Hongjian Li, Jun Chen.

**Resources:** Chuan Xia, Hongjian Li, Jun Chen.

**Supervision:** Chuan Xia, Hongjian Li, Jun Chen.

**Validation:** Ruilin Li, Sisi Xia, Xin Li.

**Writing – original draft:** Ruilin Li, Sisi Xia, Xin Li.

**Writing – review & editing:** Chuan Xia, Hongjian Li, Jun Chen.

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
