## [Decision Letter · Decision Letter 0]

28 Jul 2025

HCMV encoded UL84 hijacks FHL2 to suppress type I Interferon production and enhance viral replication

PLOS Pathogens

Dear Dr. Chen,

Thank you for submitting your manuscript to PLOS Pathogens. After careful consideration, reviewers brought forward a considerable number of concerns, so we feel that it has merit but does not fully meet PLOS Pathogens's publication criteria as it currently stands. Therefore, we invite you to submit a carefully revised version of the manuscript with an itemized rebuttal that addresses all points raised in this initial review.

Please submit your revised manuscript within 60 days Sep 26 2025 11:59PM. If you will need more time than this to complete your revisions, please reply to this message or contact the journal office at plospathogens@plos.org. Please include the following items when submitting your revised manuscript:

We look forward to receiving your revised manuscript.

Kind regards,

Edward S. Mocarski

Academic Editor

PLOS Pathogens

Robert Kalejta

Section Editor

PLOS Pathogens

Editor-in-Chief

PLOS Pathogens

orcid.org/0000-0003-2946-9497

Editor-in-Chief

PLOS Pathogens

orcid.org/0000-0002-7699-2064

**Journal Requirements:**

At this stage, the following Authors/Authors require contributions: Ruilin Li, Sisi Xia, Xin Li, Ying Zeng, Tianqi Wang, Chuan Xia, Hongjian Li, and Jun Chen. Please ensure that the full contributions of each author are acknowledged in the "Add/Edit/Remove Authors" section of our submission form.

3) In the online submission form, you indicated that "All data in the document are available by contacting the corresponding authors.". All PLOS journals now require all data underlying the findings described in their manuscript to be freely available to other researchers, either

1. In a public repository

2. Within the manuscript itself

3. Uploaded as supplementary information.

4) Please amend your detailed Financial Disclosure statement. This is published with the article. It must therefore be completed in full sentences and contain the exact wording you wish to be published.

1) State what role the funders took in the study. If the funders had no role in your study, please state: "The funders had no role in study design, data collection and analysis, decision to publish, or preparation of the manuscript.".

**Reviewers' Comments:**

Reviewer's Responses to Questions

**Part I - Summary**

Reviewer #1: Li et al provide a very nicely written, clear, interesting study of UL84 and FHL2 that is entirely suitable for publication in PLoS Pathogens. They show that FHL2 plays a novel architectural role in enhancing IFNb transcription upon infection with human cytomegalovirus or a number of other viruses, and provide detailed mechanistic analysis of how this occurs. Furthermore, they find that the HCMV UL84 protein interacts with FHL2 to help evade IFN signalling.

Reviewer #2: This manuscript by Ruilin et al. describes the role of HCMV-encoded UL84 as a negative modulator of the PRR-induced type I IFN response and identifies the FHL2 protein as a UL84 interacting protein, whose activity at the IFNbeta enhanceosome is hampered by UL84.

The interaction between UL84 and FHL2, identified in a Y2H assay, was confirmed with overexpression experiments via Co-IP in HEK293T and HFF cells, as well as confirmed by GST pulldown experiments, confirming a direct interaction. The authors further show that this interaction hampers assembly of the preinitiation complex at the Ifnb1 enhanceosome. They also show that FHL2 gets phosphorylated upon HCMV infection, which enhances Ifnb1 enhanceosome formation. While the authors present a lot of data characterising the interaction between UL84 and FHL2, mostly using overexpression studies, evidence showing the role of UL84-mediated FHL2 function during the IRF3-mediated type I IFN response in the HCMV infection context is missing. Most experiments are performed in presence of overexpressed (plasmid-derived) UL84, but reagents and methodologies do exist and are feasible to apply to show the relevance of this interaction in a more physiological context. in addition, several controls need to be included to show specificity for the observed phenotype.

Reviewer #3: I read with interest the manuscript by Li et al. “HCMV encoded UL84 hijacks 1 FHL2 to suppress type I Interferon production and enhance viral replication”. In this article, the authors performed an impressive series of co-immunoprecipitations to identify novel functions for the host gene FHL2 and the HCMV gene UL84 in the regulation of IFNbeta expression. The most notable results are that (i) the authors show that FHL2 allows the transcriptional activation of IFNbeta by bridging the TATA binding protein with the IRF3-cJun-p300 complex at the IFNbeta promoter and (ii) that they show that UL84 disrupts the FHL2 binding to the abovementioned complex by interfering with FHL2 phosphorylation at the position T112. The authors show that UL84 overexpression enhances IFNbeta in cells infected with HCMV and that it affects viral replication. They also show that UL84 binding to FHL2 occurs at nucleotide positions 95-158 on the UL84 side and 400-460 on the FHL2 side. Finally, the authors state that the

The study provides a plethora of highly convincing biochemical data on the molecular interactions of FHL2 with host target proteins and DNA sequences in the IFNbeta promoter, as well as on the interactions with UL84. Hence, these data are well worth of publishing in general. The authors are less convincing in the virological interpretation of their findings. While UL84 is encoded and expressed by HCMV, the authors typically used experimental settings where cells were transfected with UL84 and then infected. Since almost all evidence in this article depended on the extrinsic addition of UL84 prior to virus infection, which is not the natural condition, it is unclear if UL84 affects IFNbeta expression in the context of virus infection. In fact, the only assay where a more natural setting was used is shown in panels 9A+9B and they show that the RNAi against FHL2 decreased the IFN responses and increased virus titers, although UL84 was present in this system. However, the manuscript title states that UL84 interaction with FHL2 enhances virus replication by decreasing IFNbeta. In light of my concerns, the effect of UL84-FHL2 interaction on virus replication seems insufficiently supported by hard evidence.

**Part II – Major Issues: Key Experiments Required for Acceptance**

Reviewer #1: I have only relatively minor comments that need addressing prior to publication:

1. The authors say they ‘constructed a series of viral protein expression vectors and screened them using reporter assays for their abilities to regulate IFN-β promoter activity in HCMV-stimulated HFF cells’ – they should show this data

2. I’m not sure I agree with the statement UL84 regulated IFN in a time-dependent manner; was there a significant difference in the IFNb-luc between time points (or the fold change between HCMV+vector vs HCMV+UL84) in Fig 1b?

3. Initial experiments all use UL84 overexpression – one or two of these effects ought to be confirmed either using a UL84 deletion virus (if it is possible to generate one and complement it’s function using UL84-expressing cells) – OR using the efficient knockdown protocol they have in figure 7.

4. Authors should show the full data for the Y2H study, not just four ‘hits’. Also, where’s the confirmation of KPNA3 interaction with UL84 that is stated in Table S1. Should be shown.

5. Why was EZH2 chosen as a control for the co-IP? Should say why.

6. Quality of the IF images in 2C was too poor to assess properly. Should show more cells, and potentially quantify the co-expression. In general the image quality was grainy – I appreciate this was only an initial submission, however high quality images must be provided for publication.

7. How is the effect on IFIT1 mediated? It is known to be directly responsive to IRF3 (i.e. prior to expression of IFN) – is this regulated similarly to IFNb? How about ISG15?

8. “The results indicate that virus infection induces the phosphorylation and nuclear translocation of FHL2, which may serve as a cellular sensor-like role for detecting viral invasion” – or could it just be phosphorylated as part of a signalling cascade after sensing via known pathways? Similarly, the statement “function as a cellular sensor-like role in viral invasion” in line 427 and “we demonstrated that phosphorylation of FHL2 serves as a virus-triggered sensor-like role” (line 464) should be revised – they have not shown that FHL2 has a sensing role, more that it is involved in promoting IFNb transcription.

9. The two statements “These results indicate FHL2 is a crucial mediator for virus-induced transcriptional activation of type I interferon” and “Considering FHL2 is proven to be a key regulator in the mechanism of IFN-β transcription” use results almost all determined earlier in the paper by FHL2 overexpression, apart from Figure 4D-E where the effect of si knock-down of FHL2 on STAT1 phosphorylation is weak. To confirm importance, they should knock it down our out in other experiments. They do a nice FHL2 knock-down for Figure 6B (or knock-out – should be clarified?? I suspect this was a si knock-down – should be stated as such. If knockout, provide methods), so should use this strategy in earlier sections of the paper to look at interferon regulation.

10. They should quantify all blots, at least for the supplementary, and state the number of replicates of every experiment in the figure legend.

Reviewer #2: Major points:

1. a) Figure 1: The luciferase reporter assays are conducted in HFF cells that were transfected with reporter plasmids plus empty vector or UL84, and then infected with HCMV with an MOI of 1, meaning that every cell must have gotten infected. Expression of UL84 upon transfection was quite strong and I am surprised to see that transfection has worked so efficiently. Can the authors please explain how they reached such high transfection efficiency? While I see that only cells can respond with an IFN response that received the reporter plasmids plus UL84, and not untransfected cells, this is not the case for HFF cells that were transfected, infected (MOI 1) and analysed with RT-PCR experiments for Ifnb1 (panel C). Such a strong phenotype can only be reached if most cells would have been transfected with UL84 - and I doubt that this can be achieved in HFF.

b) While UL84 also seems to inhibit the PRR-mediated IFN response upon IAV, EV71, HSV and ZIKV infection, it would be nice to include a control such as TNFalpha or other pro-inflammatory cytokines, which should not be affected (NF-kB reporter assay, RT-PCR). What could also be tested would be IFNAR signaling - this should not be affected by UL84 if it acts via FHL2 (IFNbeta stimulation, RT-PCR for IRF9).

c) Panel A: Why can UL84 not be detected in HCMV infected cells (column 2)? According to the IP experiment in Figure 2 panel D, it should be expressed 24h post infection.

2. The IRF3-mediated IFN response is induced shortly after infection. I find no proof in the manuscript that UL84 is expressed at this time point. A kinetic experiment should be shown analysing UL84 expression at different time points post transfection, together with p-IRF3 IBs.

3. Figure 2: The IF shown in panel C should be conducted with endogenous FHL2 instead of overexpressed FHL2.

4. If UL84 indeed inhibits the IFN response, cells transfected with UL84 should show lower titers of IAV. This would be an experiment that would convincingly show that it has a biological effect.

5. Figure 4 A, lane 2: why is endogenous FHL2 expression not induced upon HCMV infection (24h p.i.)? I would suggest to do this assay at the time point when it was shown to be induced, 4-6 hpi (Figure 3). As shown in panel B of this figure, the IFN response is highest 6 h pi and declines afterwards. For panel B, it would be important to include an IB control showing endogenous UL84 expression and include an FHL2 and GAPDH IB.

6. Figure 4 D+E: siRNA transfection targeting FHL2 has no effect on p-STAT levels. This is not mentioned in the result section. The same is true for overexpression of FHL2 - no effect on p-STAT levels. However, here the authors claim that p-STAT levels are affected, with which I disagree with. While P-IRF3 levels should not be affected, p-STAT levels should be affected of FHL2 is important for type I IFN induction. This experiment weakens the study significantly and it is irritating that the data is interpreted entirely differently by the authors.

7. Figure 5 C: is not labeled comprehensively. Lacks an EV control (no FHL2 control).

8. ChIP assay: I do not think one can conclude that FHL2 binds to the region -84--+46 based on this experiment. This figure would benefit from a comprehensive scheme of the Ifnb1 promoter/enhancer, the IRF3 etc. binding sites and the primer binding sites.

9. Figure 7: Panel A should be labeld more clearly (are all lanes HCMV-infected?). The control FLH2 + EV is missing (as control for FHL2 + UL84).

10. Figure 7: Since the UL84 siRNAworks well, this experiment should be conducted with UL84 expressed in the virus context. Without UL84 plasmid-derived overexpression. As control, pIRF3 (should not change) and pSTAT (should be affected) IBs should be done.

11. The authors use HCMV strain TOWN throughout their study. Since UL84 is esssential for viral replication, a knockout virus could not be tested and experiments were conducted with overexpression of UL84 or siRNA targeting UL84. However, HCMV strain TB40 can replicate without UL84 and hence it would be possible to test whether HCMV lacking UL84 induces a stronger IFN response compared to WT virus.

12. Figure 8 panel A lacks the control HCMV + EV.

13. Figure 9 lacks IB controls which would be essential to validate the presented results.

14. Manuscript lacks information about the number of biological and technical replicates.

Reviewer #3: 1. Show effects of UL84 in the context of virus infection. A virus mutant with a full deletion of UL84 may is not easy to generate due to its essential requirement for viral DNA replication, but it may be grown o na complementing cell line. It may also be possible to generate a mutant virus where UL84 can still act with UL44 on virus DNA replication but is unable to bind to FHL2. The authors show in figure 2 that a deletion of nucleotide positions 95-158 abrogates its binding to FHL2. Hence, a mutant virus lacking only these nucleotides may be viable, which would allow the identification of UL84 effects on the FHL2-IFNbeta signaling axis.

2. It is also possible that UL84 expression during the lytic virus cycle occurs too late to affect IFNbeta responses. This would invalidate a major claim from the title of the manuscript. Hence, it may be interesting to explore the effects of UL84 in models of HCMV latency and reactivation, where UL84, if expressed early during reactivation, may affect IFNbeta responses and facilitate the full blown virus reactivation. At a minimum, these items need to be discussed as alternative scenarios, while the title is changed to reflect more accurately the available evidence provided in the manuscript.

**Part III – Minor Issues: Editorial and Data Presentation Modifications**

Reviewer #1: Minor comments

Line 270 ‘enhancersome’ spelled incorrectly

‘did not be co-immunoprecipitated with TBP’ -> ‘was not co-immunoprecipitated…’

Reviewer #2: Other comments:

1. The information provided in the M+M section is not sufficient to recapitulate the experiments in most sections:

a) The origin of the cell lines is not provided (HFF and 293-T). Ideally, also the passage number should be provided.

b) Details on the transfection of HFF cells are not provided. HFF are difficult to transfect and I do not know any lab that has schieved it to a level as high as shown here. It would be great to share the protocol with the scientific community. How many microgramms of DNA were transfected per well? This should also be stated for the titration experiments.

c) Information on the luciferase plasmid is not sufficient. Which sequence of the IFNb1 promoter/enhancer is inserted upstream of the luciferase gene?

d) Information on the UL84 plasmid is not sufficient. Is it UL84 derived from TOWN?

e) The reference ID of the TOWN BAC needs to be provided. Was the TOWN-BAC verified by sequencing?

f) Some IPs and subsequent immunoblots were performed with the same antibody and are impressively clean. It would be good to state in the M+M section which antibody was used for which application and also mention the species it was made in (IP, IB).

2. I was intrigued to see the molecular weigth difference of unphosphorylated FHL2 (32 kDa) and its phosphorylated version which runs as high as 95 kDa (Figure 3 panel G-K). Can the authors please explain this phenomenon?

3. Did the authors do an alphafold prediction with UL84 and FHL2 to analyse their interaction?

4. The depth of information about the Y2H assay are scarce and the Y2H assay results are currently only listed in a supplementary table. More detail (in the method section as well as the result section) should be shown. Ideally, results of the Y2H assay should be shown as figure 1 or 2.

5. In their introduction, the authors could elaborate more about UL84.

Minor comments (language related):

1. abstract: "...binds to the cellular receptor..." - to what are the authors refering to here?

2. Line 66 and onwards: The authors cite >20 year old papers regarding the IFNbeta enhanceosome. While it is important to cite original papers and give credit to those authors, a more recent review article on IRF3 could be cited as well (for example Schwanke et al 2020 doi: 10.3390/v12070733).

3. Line 60: ..."upon the viral binding of PRRs..." is phrased in a strange manner - the virus itself does not bind to PRRs, rather viral ligands or PAMPs.

4. Line 97: please write out for what LIM stands for

5. Line 118: " HCMV is the largest herpesvirus in the genome..." needs to be rephrased.

6. Line 582: typo (Reagents)

7. Line 665: strange phrasing ("there are six main concerns...")

8. Line 703-704: strange sentence

9. Figure 2 panel E and G should be labeled more clearly. Panel D: MW marker is missing.

10. Line 270: enhanceosome

Reviewer #3: 1. There is very little information provided about the plasmids pcDNA3.1-UL84 and pGL3-IFN-beta. Wat promoter is used in the former one? Is the latter one encoding the IFNbeta promoter? Which part of the promoter? Does it also express IFNbeta or just the luciferase? What is the evidence that it matches the patter of IFNbeta gene expression at the native site? If there is a previous publication about these plasmids, it needs to be more clearly cited. If they are new constructs, more details on their characterization need to be provided.

2. pGL3-IFN-beta is mentioned in some part of the text, but not in M&M, where the only plasmid with a similar name is the pGL3-luci-IFN-beta. Please adjust and standardize this nomenclature.

3. What MOI was used in figures 1D-1G?

4. 293T cells (HEK 293 T cells?) are used for cotransfections of UL84 and FHL2. It would be useful to state once they are introduced for the first time if these cells express FHL2 at low levels or not at all.

5. Please indicate molecular weights in figure 2D

6. Lines 301 state that indicate that “all four predicted sites (Y97, S98, S99, and T112) within the LIM2 domain are essential for FHL2’s interaction with IRF3, c-Jun, or p300 during the assembly of the IFN-β enhanceosome complex.” However, it may also be possible that some or all of these sites are important for correct FHL2 protein folding, rather than a direct interaction with the target proteins.

7. Figures S4 and S5 seem to be swapped in the text.

8. In figure 6B, there is still a band for TBP, though it is less pronounced. A similar reduction in signal is observed for Jun and p300. So, the text in lines 318-320 (“when we knocked out the endogenous FHL2, IRF3 failed to be co-immunoprecipitated with TBP, although it still recruited p300 and c-Jun to the IFN-β promoter”) does not accurately describe the figure 6B.

9. The text in lines 320-21 “Molecular docking of complex form the IRF3-FHL2-TBP demonstrates that IRF3 specifically recognizes the N-terminal LIM domain of FHL2,” is confusing, please rephrase.

10. In figure 7H, it appears that an IP was performed on UL84, but the text states that it was done on FHL2. If the IP was on FHL2, how comes the band for FHL is absent in the WB in the second row, second column?

11. How comes HCMV infection induces IFNbeta in the second column of figure 8A, even though HCMV encodes the UL84 gene?

12. While the PKC inhibitor LY33531 in figure 8F shows an effect on nuclear translocation of FHL2, this experiment would be corroborated had the authors used a PKC-specific siRNA.

13. The data in figure 9B should be shown as growth curves on a logarithmic scale, not using a linear scale. Virus growth is exponential so should be the scale that display it.

14. The figure 9C is confusing – they say that they measure OryLit replicative activity by using a luciferase reporter, which is indicative of transcription and gene expression, not of DNA replication. Please explain this better. This growth curve should be performed more ideally over a period of a week, not only for 4 days, because HCMV replicates slowly in cell culture.

15. Numerous mistakes in sentence structure were observed in lines 99, 103, 106, 109, 118 119-120, 317, 320, 359, 380, 381, …

PLOS authors have the option to publish the peer review history of their article (what does this mean? ). If published, this will include your full peer review and any attached files.

**Do you want your identity to be public for this peer review?** For information about this choice, including consent withdrawal, please see our Privacy Policy .

Reviewer #1: No

Reviewer #2: No

Reviewer #3: No

**Figure resubmission:**

**Reproducibility:**



---

## [Editor Report · Decision Letter 1]

13 Jan 2026

Dear Dr. Chen,

We are pleased to inform you that your manuscript 'HCMV encoded UL84 hijacks FHL2 to suppress type I Interferon production and enhance viral replication' has been provisionally accepted for publication in PLOS Pathogens.

Best regards,

Edward S. Mocarski

Academic Editor

PLOS Pathogens

Robert Kalejta

Section Editor

PLOS Pathogens

Sumita Bhaduri-McIntosh

Editor-in-Chief

PLOS Pathogens

orcid.org/0000-0003-2946-9497

Michael Malim

Editor-in-Chief

PLOS Pathogens

orcid.org/0000-0002-7699-2064

Your revised manuscript has been reviewed and found to address a majority of concerns raised by reviewers.
---

## [Editor Report · Acceptance letter]

Dear Dr. Chen,

We are delighted to inform you that your manuscript, "HCMV encoded UL84 hijacks FHL2 to suppress type I Interferon production and enhance viral replication," has been formally accepted for publication in PLOS Pathogens.

Best regards,

Sumita Bhaduri-McIntosh

Editor-in-Chief

PLOS Pathogens

orcid.org/0000-0003-2946-9497

Michael Malim

Editor-in-Chief

PLOS Pathogens

orcid.org/0000-0002-7699-2064